# Synthesis of a magnetic π-extended carbon nanosolenoid with Riemann surfaces

Jinyi Wang [1,6], Yihan Zhu [2,6], Guilin Zhuang [3,6], Yayu Wu[1], Shengda Wang[1], Pingsen Huang[1], Guan Sheng[2], Muqing Chen[1], Shangfeng Yang [1], Thomas Greber [4] & Pingwu Du [1,5✉]

Riemann surfaces are deformed versions of the complex plane in mathematics. Locally they look like patches of the complex plane, but globally, the topology may deviate from a plane. Nanostructured graphitic carbon materials resembling a Riemann surface with helicoid topology are predicted to have interesting electronic and photonic properties. However, fabrication of such processable and large π-extended nanographene systems has remained a major challenge. Here, we report a bottom-up synthesis of a metal-free carbon nanosolenoid (CNS) material with a low optical bandgap of 1.97 eV. The synthesis procedure is rapid and possible on the gram scale. The helical molecular structure of CNS can be observed by direct low-dose high-resolution imaging, using integrated differential phase contrast scanning transmission electron microscopy. Magnetic susceptibility measurements show para-magnetism with a high spin density for CNS. Such a π-conjugated CNS allows for the detailed study of its physical properties and may form the base of the development of electronic and spintronic devices containing CNS species.

[1] Hefei National Laboratory for Physical Sciences at the Microscale, iChEM (Collaborative Innovation Center of Chemistry for Energy Materials), CAS Key Laboratory of Materials for Energy Conversion, Department of Materials Science and Engineering, University of Science and Technology of China, 96 Jinzhai Road, Hefei, Anhui Province 230026, China. [2] Center for Electron Microscopy, State Key Laboratory Breeding Base of Green Chemistry Synthesis Technology and College of Chemical Engineering, Zhejiang University of Technology, Hangzhou 310014, China. [3] College of Chemical Engineering, Zhejiang University of Technology, 18 Chaowang Road, Hangzhou, Zhejiang Province 310032, China. [4] Physik-Institut, University of Zürich, Winterthurerstrasse 190, CH-8057 Zürich, Switzerland. [5] National Synchrotron Radiation Laboratory, University of Science and Technology of China, 42 Hezuohua South Road, Hefei, Anhui Province 230029, China. [6] These authors contributed equally: Jinyi Wang, Yihan Zhu, Guilin Zhuang. ✉email: dupingwu@ustc.edu.cn

As a two-dimensional (2D) $sp^2$ carbon allotrope that consists of only a single layer of carbon atoms arranged in a honeycomb lattice, graphene (Fig. 1a) and graphene-based nanomaterials have captured enormous attention and interest in the past two decades[1–5]. The single-layer and few-layered graphene materials exhibit many attractive properties such as large specific surface area, excellent chemical and thermal stability, high conductivity, and high charge carrier mobility[3,6]. Thus, many graphene-based applications have been reported in organic electronics, composites, sensors, spintronics, optoelectronics, and photonics[2,4,7]. Semiconducting graphene-based nanostructures with a finite bandgap show interesting electronic properties that do not exist in extended graphene[8–10]. For instance, the edge states of zigzag graphene nanoribbons (ZGNRs, Fig. 1b) are expected to be ferromagnetically coupled along the edge and anti-ferromagnetically coupled between the edges[11]. Nanostructured graphitic carbons with zigzag edges are predicted to possess spin-polarized electronic edge states and can thus play important roles in graphene-based spintronics[11,12]. However, owing to the limited accuracy of current top-down methods, direct observations of spin-polarized edge states of ZGNRs are limited[9,13–16]. Recently, a few small nanographene-based examples were synthesized by on-surface synthesis and demonstrated magnetic properties[17,18]. The quantum confinement in armchair graphene nanoribbons (AGNRs, see Fig. 1b) and carbon nanotubes (CNTs) results in the opening of substantial electronic bandgaps that are directly related to their structural boundary conditions[19–24].

In three-dimensional (3D) graphene structures, some interesting topology like helical spirals from graphite screw dislocations has been proposed[25]. Moreover, four kinds of dislocations and helical shapes were observed in raw anthracite using the bright-field high-resolution transmission electron microscopy (HRTEM)[26]. For the present case of nanosolenoids, one atomic graphene plane continuously spirals around the line perpendicular to the basal plane, which can be considered to closely follow a Riemann surface

(namely, a log z type). As well-known objects in mathematics, Riemann surfaces (Fig. 1c and Supplementary Fig. 1, representative examples of Riemann surfaces) were proposed by Riemann in 1851 to predict a single-valued domain for a multivalued analytical function. It is noteworthy that the Riemann surfaces not only play key roles in the development of modern mathematics but also provide insights for the design and synthesis of multifunctional curved carbon materials[25–28]. In 2016, Yakobson and coworkers initially predicted that a carbon solenoid with Riemann surfaces and small diameter can behave as a quantum conductor when a voltage is applied, resulting in a large magnetic field near the center and bringing about excellent inductance[25].

Previous experimental and theoretical studies have explained that folding graphene causes changes in the material characteristics and can be used to tune its electronic[29] and photophysical properties[30]. Thus, the design and synthesis of curved large π-extended graphene-based nanomaterials have attracted great research interest in the quest to develop carbon nanostructures with distinctive geometric shapes and optoelectronic properties. For example, a series of curved carbon molecules have been successfully synthesized, in which saddle-shaped geometries formed due to the presence of five-, seven-, or eight-membered-ring defects[28,30–37] or steric hindrance[38,39]. Some of these curved nanographenes have demonstrated interesting physical properties in organic field-effect transistors (OFET)[31,32,36], near-infrared absorption and fluorescence[40,41], two-photon absorption[28,35,37], electronic circular dichroism (ECD)[28,35], and circularly polarized luminescence (CPL)[42,43].

In this work, we report a facile bottom-up synthesis of a 3D π-extended nanographene carbon material CNS that mimics a Riemann surface with high yields, as shown in Fig. 1d-I and Supplementary Fig. 2. Notably, the present CNS material has a low optical bandgap, strong red photoluminescence, and complex slow magnetic ordering behavior at low temperatures. This helical carbon material offers a chance for the investigation of its physical and magnetic properties.

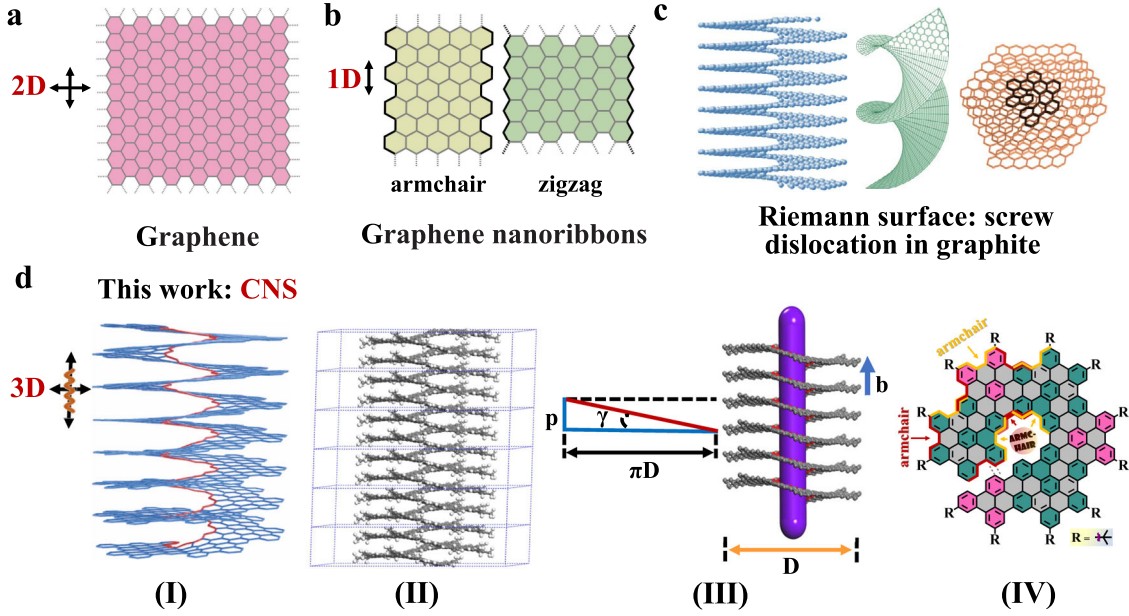

**Fig. 1 Schematic illustrations of known or predicted carbon allotropes. a** 2D extended graphene, the thinnest $sp^2$ allotrope of carbon arranged in a honeycomb lattice. **b** 1D graphene nanoribbons with armchair edges (AGNRs) and zigzag edges (ZGNRs). **c** Examples of the predicted structure of helically twisted graphenes with Riemann surface. **d** (I) Design of the 3D fully π-extended curved single-stranded carbon nanosolenoids (CNS)[25]. (II) The atomic structure of CNS with side view by the DFT calculations. (III) Helical structure of CNS without the alkyl substituent groups. Burgers vector **b** parallel to the c-axis (purple). **D**, **p**, and **γ** are the outer diameter, pitch, and coil angle, respectively. (IV) Edges and helical core structures of CNS. Fig. 1c (left) adapted with permission from ref. [25] (copyright 2016 American Chemical Society), Fig. 1c (middle) adapted with permission from ref. [66] (copyright 2019 American Physical Society), and Fig. 1c (right) adapted with permission from ref. [67] (copyright 2018 American Chemical Society).

## Results

**Molecular design and theoretical calculations of CNS.** Figure 1c shows the conceptual structure of nanostructured graphitic carbon materials resembling a Riemann surface with helicoid topology. In order to obtain the target graphene-based material with the desired structural features, including large π-conjugation length, single-stranded structure, and specific edge configuration, it is important to select the appropriate building units rationally. As the fundamental structure of polycyclic aromatic hydrocarbons (PAHs) and an important segment of 2D graphene materials, a hexaphenylbenzene (HPB) derivative with two boryl groups (**M1**, Fig. 2) was chosen as the PAH building unit to achieve the large π-extended feature of CNS. The phenanthrene moiety (colored green) in **M1** can increase the rigidity and steric constraint of the building unit. Besides, another building unit **M2** was used as the linker with two ortho-bromo groups (colored red) to further increase the steric constraint and achieve the helicoid topology of CNS and two ortho-diphenyl groups (colored yellow) to elongate the acenes and allow annulation of benzene rings. To investigate atomic structure information such as the diameter, helical pitch, edges, helical core, coil angle, stacking, band, spin density, molecular orbitals, and so on, density-functional theory (DFT) calculations were carried out for the geometric structure and electronic structure of CNS using the Vienna an initio simulation package (VASP). The results are summarized in Fig. 1d (II and III), Supplementary Figs. 29–30 and Supplementary Table 1. Figure 1d-II shows the geometrical optimization structure of CNS, and the results indicate that CNS features helix-bundles configuration along the Burgers vector **b** (parallel to the c-axis marked in purple, see Fig. 1d-III) with the outer diameter,

helical pitch, and coil angle of $D = 24.782$ Å, $p = 4.118$ Å, and $\gamma = 3.028°$ [$\gamma = \arctan(p/(\pi D))$], respectively[25]. Furthermore, molecular dynamic simulation results at 300 K also reveal the thermal stability of the bundles configuration (supplementary movie). CNS is a carbon material that has armchair edges and armchair helical core structures (marked in Fig. 1d-IV).

**Solution-processable synthesis of CNS.** The key synthesis procedure for CNS and its polyphenylene precursor (**P1**) are shown in Fig. 2 and Supplementary Figs. 3, 4. Briefly, it is based on a polymerization of the precursor **P1** by a Pd-mediated Suzuki coupling followed by a Scholl reaction for cyclodehydrogenation to form CNS. The characterization data of nuclear magnetic resonance (NMR) and HR-MS (MALDI-TOF) and the synthesis details for intermediate compounds are available in the Supplementary Information (Supplementary Figs. 14–28). The polymerization of **M1** and **M2** was performed by a Suzuki coupling reaction using Pd(PPh₃)₄ as the catalyst and Aliquat 336 as the phase transfer catalyst. The reaction was run at 110 °C for 5 days in a mixed solvent of toluene/H₂O (v/v, 5:1) to obtain the polyphenylene precursor **P1** as a pale white solid. Oxidative cyclodehydrogenation of **P1** was performed via the Scholl reaction using 2,3-Dichloro-5,6-dicyano-1,4-benzoquinone (DDQ) and trifluoromethanesulfonic acid (TfOH) at 0 °C for 15 h, resulting in the final product of CNS as a dark black solid (Fig. 4b). This solid emits strong red fluorescence in a diluted solution, indicative of significant structural changes in the π-conjugated backbone. The weight average molecular weight ($M_W$), relative number-average molecular weight ($M_n$), and polydispersity index (PDI) of **P1** were estimated by gel permeation chromatography

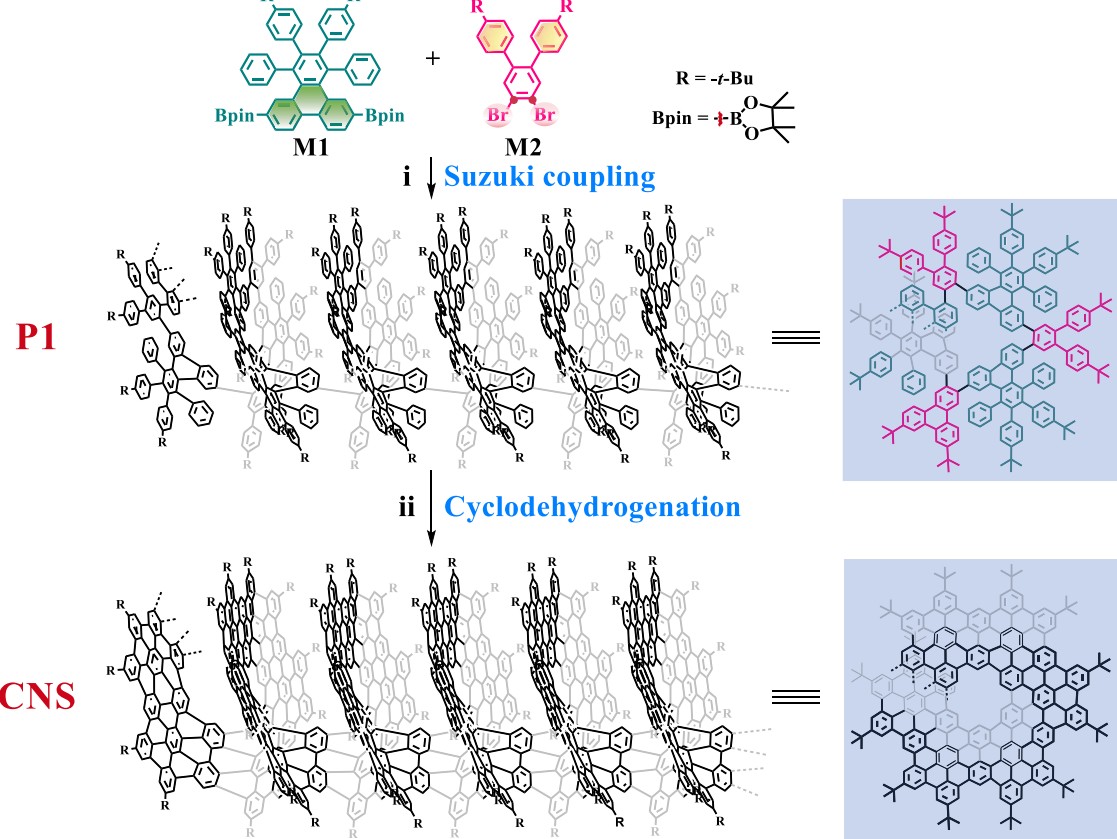

**Fig. 2 Synthesis approach to the 3D fully π-extended curved single-stranded CNS.** Reagents and conditions: (i) **M1** (1.0 equiv.), **M2** (1.0 equiv.), K₂CO₃ (10 equiv.), Aliquat 336 (5 mol%), Pd(PPh₃)₄ (10 mol%), Ar, toluene/H₂O (v/v, 5:1), 110 °C, 5 days; (ii) **P1** (1.0 equiv.), DDQ (14 equiv.), TfOH, Ar, anhydrous CH₂Cl₂, 0 °C, 15 h.

(GPC) (Supplementary Fig. 5). The molecular weight distribution of P1 shows a single broad peak with a low PDI of 1.34, and the $M_n$ is 27100 g·mol$^{-1}$, corresponding to a degree of polymerization of 37 units. Considering that one helical patch consists of three monomeric units, the $M_W = 36300$ g·mol$^{-1}$ of P1 represents ~12 helical patches on average.

**Characterization of CNS and P1.** The high efficiency of graphitization of precursor P1 into CNS was evaluated by Fourier transform infrared (FT-IR), Raman, and solid-state NMR measurements. As shown in Fig. 3a and Supplementary Fig. 6, solid-state NMR spectra were performed to confirm the structural changes before and after the oxidative cyclodehydrogenation (Scholl) reaction. The solid-state $^{13}$C NMR spectrum for

precursor P1 showed two groups of carbon resonances: the characteristic singlets ($\delta = 25.60$ to 36.31 ppm) can be assigned to $t$-butyl carbons and the other multiple peaks ($\delta = 115.70$ to 152.40 ppm) should be from aromatic carbons. In the aromatic carbon region, the broad peak at 138 ppm is from the ipso carbons in aromatic ring-$sp^2$ hybridized carbons attached to other carbons, while the broad peak at 130 ppm is from $sp^2$ hybridized carbons attached to protons. The main aromatic $^{13}$C NMR signals of CNS merged together (centered at ~121 ppm), probably because most of the carbon atoms in the CNS are bonded to other carbon atoms. Moreover, the peaks of CNS in the aromatic zone move slightly to the low field as the degree of π-conjugation increases.

Aromatic protons were significantly reduced and the $^1$H line width significantly increased by comparing the $^1$H NMR magic-

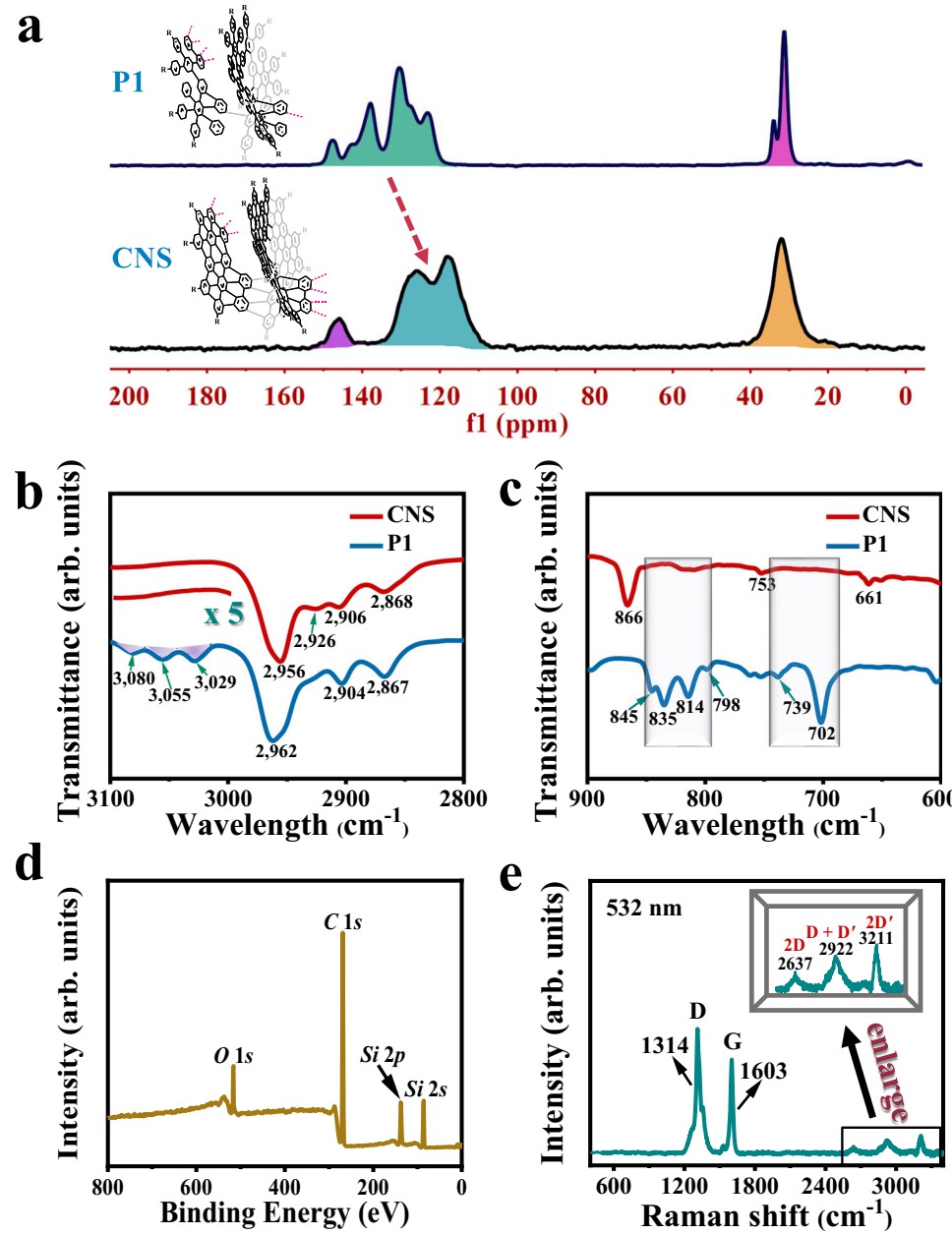

**Fig. 3 Structural characterizations of CNS. a** Solid-state $^{13}$C NMR spectra of **P1** and CNS. **b, c** Representative FT-IR spectra regions of polyphenylene precursor **P1** (blue lines) and CNS (red lines) show the disappearance of the bands derived from mono- and disubstituted benzene rings after graphitization. **d** XPS spectrum of CNS. **e** Raman spectrum of CNS measured at 532 nm (2.33 eV) on a powder sample with laser power below 0.1 mW. The inset shows a magnified area of the spectrum (bottom right) to display peaks of 2D, D + D′, and 2D′.

angle spinning (MAS) spectra of **P1** and CNS, indicating that **P1** with a semi-flexible to semi-rigid structure becomes rigid after graphitization to CNS (Supplementary Fig. 6a). Moreover, by comparing the solid-state 2D $^1$H-$^1$H double-quantum single-quantum (DQ-SQ) NMR correlation spectra of **P1** and CNS (see Supplementary Fig. 6b, c), the results show that the $^1$H-$^1$H auto-correlation signals of the aromatic protons in **P1** disappeared after graphitization to CNS and the aromatic protons in CNS are far away from each other, which is consistent with the structural characteristics of CNS. The $^1$H NMR signals of bulk CNS are significantly broadened, possibly due to the heterogeneous packing of CNS, and the currents of aromatic/anti-aromatic rings cause the $^1$H NMR signals to shift in the opposite direction[8].

Figures 3b, c show the FT-IR spectra of CNS and **P1**. FT-IR analysis of precursor **P1** and CNS, before and after graphitization, revealed the disappearance of out-of-plane C-H deformation bands located at 702, 739, 798, 814, 835, and 845 cm$^{-1}$, which are typical for mono- and disubstituted benzene rings. The aromatic C-H stretching vibrations modes located at 3029, 3055, and 3080 cm$^{-1}$ were significantly weakened as fewer aromatic C-H groups are present in CNS. In addition, the out-of-plane band typical for aromatic C-H at the cove position appeared at 866 cm$^{-1}$, which further confirmed the efficient conversion of precursor **P1** into CNS. The intensity of the signals associated with the C-H stretching of alkyl chains changed slightly (2906 and 2868 cm$^{-1}$ for CNS, 2904 and 2867 cm$^{-1}$ for **P1**) upon graphitization, indicating the integrity of the alkyl substituents. The elemental composition of CNS was detected by X-ray photoelectron spectroscopy (XPS) (Fig. 3d). The survey scan revealed only the peaks associated with the CNS and the Si substrate. The C 1 s signal is located at 269 eV with a single sharp peak corresponding to the $sp^2$ carbons. No other peaks are observed from the carbon species in different oxygen-containing functionalities[44], further confirming that this CNS material is chemically pure and not oxidized in air.

Figure 3e demonstrates a Raman spectrum of CNS (powder sample) before and after graphitization. The result shows an apparent fine structure with characteristic intense G and D bands, which are typical features for nanostructured GNRs[22,45]. The G and D peaks are located at ~1603 and ~1314 cm$^{-1}$, respectively. Due to the quantum confinement that relaxes the Raman selection rule, the G peak has a large full width at half maximum (~25 cm$^{-1}$). The intense D peak of CNS is activated by the confinement of $\pi$-electrons into a finite-size domain and is consistent with previous studies of large $\pi$-extended polycyclic aromatic hydrocarbons[46,47], which can be explained by collective modes of the confined hexagonal rings[48]. The relative intensity of the G and D peaks also indicated the small amount of defect formed during the graphitization process[49]. Besides, double formants belonging to 2D, D + D′, and 2D′ bands are also observed at 2637, 2922, and 3211 cm$^{-1}$, respectively.

**Photophysical properties**. The photophysical properties of CNS were studied by ultraviolet-visible (UV-Vis) absorption spectroscopy, steady-state fluorescence spectroscopy, and time-resolved fluorescence decay spectroscopy (Fig. 4). The polymer precursor **P1** was used as a reference for comparison. The UV-Vis spectrum of precursor **P1** showed an absorption band with a maximum absorption peak at ~299 nm. In sharp contrast, the CNS demonstrates an obvious red-shifted absorption feature in the range of 250–650 nm, with two vibronic bands at 426 and 487 nm, indicating that a larger $\pi$-conjugated structure was formed after the Scholl reaction. The bandgap of GNRs sensitively depends on its structure, width, edge, etc., and the absorption onset wavelength ($\lambda_{onset}$) of CNS was observed at ~629 nm, corresponding to the optical bandgap ($E_{gap}$) is ~1.97 eV[8].

Photoluminescence (PL) spectra of **P1** and CNS were recorded with 300 and 460 nm excitation light, respectively (Fig. 4a). In THF solution, CNS exhibits strong band emission in the range of 520-860 nm maximized at 660 nm, which is significantly red shifted (>260 nm) compared to the reference **P1**. This emission redshift is consistent with the UV-Vis result and suggests a much larger $\pi$-extended conjugation nature in CNS. In a thin film, the emission band is further red shifted and the maximum peak is located at ~678 nm. As illustrated in the Fig. 4b, precursor **P1** is a white solid, while CNS is a black red solid. The dilute THF solution of CNS shows an intense red photoluminescence under irradiation by a hand-held UV lamp at $\lambda = 365$ nm, while the precursor **P1** has an intense blue fluorescent color (Fig. 4b).

The PL lifetime ($\tau_s$) of precursor **P1** and CNS were further measured by time-resolved fluorescence decay using the time-resolved photoluminescence (TRPL) technique. Interestingly, the lifetimes of precursor **P1** were ~3.2 and 9.0 ns at 400 nm by dual-exponential fitting when excited at 390 nm (Fig. 4c). As for the CNS, dual-exponential decay was observed with $\tau_s$ ~43.6 and 8.6 ns at 660 nm when excited at 560 nm (Fig. 4d). These two emission lifetimes could originate from the different excited states of a helix pattern of CNS. This observation also indicates the significant difference between the precursor **P1** and CNS with a large $\pi$-conjugation helical carbon structure.

**Structural elucidation**. Transmission electron microscopy (TEM) provides a powerful tool for visualizing the structures of organic molecules and polymers by direct imaging[50,51]. It even allows the explicit elucidation of quite complicated spiral nanostructures[52]. One major challenge for the direct TEM imaging of the CNS helix is the electron-beam irradiation induced structural damage. Unlike graphene with a two-dimensional covalent network and predominantly subjected to the knock-on damage mechanism[53], the beam damage mechanisms for helical CNS molecules are more complicated and may include considerable ionization effects. This means that simply lowering the accelerating voltage of electron beam that greatly eases the knock-on damage may not be able to alleviate the overall beam damage of CNS molecules. We evaluate the beam damage effects over CNS molecules using conventional TEM operated under a moderate accelerating voltage of 200 kV. Figure 5a shows the low-magnification TEM image covering a relatively large area of film assembled by CNS molecules. Careful inspection reveals that the CNS molecular strands assembled into domains with spring-like lattice fringes characterized for the helical structure (Fig. 5a-I and Supplementary Fig. 8), which probably originates from the magnetic interactions among CNS molecular strands. From an enlarged region of interest (Fig. 5a-II), the helical pitch of ~0.41 nm can be clearly measured from the width of periodic lattice fringes of CNS strands, which corresponds to the spots in the fast Fourier transform (FFT) pattern (Fig. 5a-III) from the white rectangle in Fig. 5a-I. The HRTEM images in Supplementary Fig. 9 show lattice fringes corresponding to XRD patterns (Supplementary Fig. 7) of self-assembly CNS, and the lattice fringe spacings are 2.65, 1.92, and 1.11 Å, respectively, probably due to the self-assembly of CNS polymer. Notably, the packing order of CNS molecules is rarely observed by low-magnification TEM or scanning electron microscopy (Supplementary Fig. 11), which is however visible from powder XRD pattern and high-magnification TEM images taken from small local regions (Supplementary Fig. 9). Accordingly, the packing order of CNS assembly is most likely damaged by electron beam irradiation and further molecular structure damage is observed upon prolonged irradiation, which prohibits the explicit structural elucidation of single-stranded CNS helices.

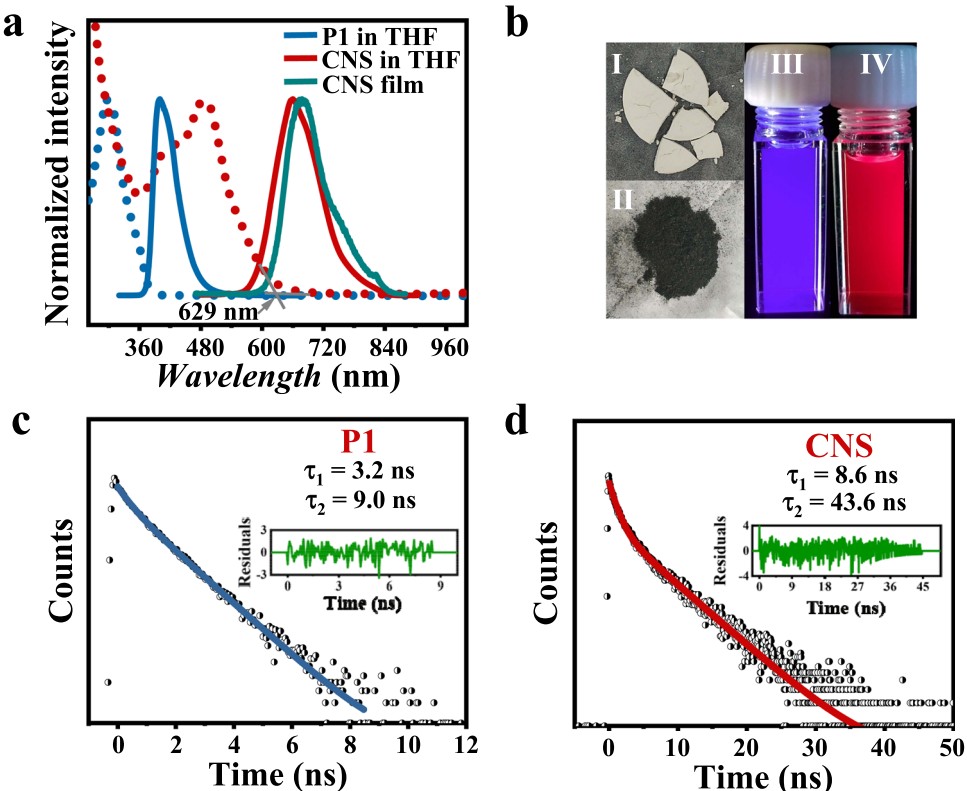

**Fig. 4 Photophysical properties of CNS. a** UV-Vis absorption (short dot lines) and fluorescence (solid lines) spectra for polyphenylene precursor **P1** (blue lines), CNS in THF (red lines), and CNS in thin film (green line). **b** Photograph showing the solid powder of polyphenylene precursor **P1** (I) and CNS (II), and the fluorescence for polyphenylene precursor **P1** (III) and CNS (IV) in THF under a UV lamp at $\lambda = 365$ nm. **c** Emission lifetime measured at 400 nm for polyphenylene precursor **P1** in THF. Inset: Curve fitting residuals for polyphenylene precursor **P1**. **d** Emission lifetime measured at 660 nm for compound CNS in THF. Inset: curve fitting residuals for compound CNS.

It has been recently reported that low-dose electron microscopy provides an efficient solution for the atomic- or molecular-resolution imaging of beam-sensitive materials[54]. As a typical low-dose imaging technique, integrated differential phase contrast scanning transmission electron microscopy (iDPC-STEM) has been proved to be quite dose-efficient and sensitive to light elements[55,56], which is very suitable for high-resolution imaging of beam-sensitive organic materials like CNS helices. The severely aggregated CNS helices are well separated after prolonged probe sonication, which are then subject to low-dose iDPC-STEM imaging for resolving intrinsic single-stranded molecular structures (Fig. 5b). Specifically, Fig. 5b–I shows the iDPC-STEM image of a single-stranded CNS helix with bright contrast that corresponds to the projected scalar electrostatic potentials of the molecule. The experimental helical pitch (**p′**) and width (**D′**) of the CNS helix can be statistical measured from the projected structure as $0.40 \pm 0.03$ nm and $2.7 \pm 0.2$ nm respectively (Fig. 5b-II and Supplementary Fig. 10), which match well with those values of the proposed CNS structural model upon a certain projection (when the alkyl sidechains were not removed, **p** and **D** were ~0.41 nm and ~2.88 nm respectively, see Fig. 5b-III). In addition, the simulated projected potential map of a single-stranded CNS structural model embedded within a 1 nm-thick amorphous carbon layer also closely resembles the high-resolution iDPC-STEM image (Fig. 5b-II). Based on all above results, the molecular structure of the CNS helices is observed by low-dose electron microscopy.

**Magnetic and electronic properties**. Previous theoretical calculations predict that such a solenoid material could have unique magnetic properties[25]. In the ${}^1$H NMR spectrum, no proton signals were observed for CNS solenoids at room temperature, prompting us to further investigate their ground-state electronic structures and possible magnetic properties. Electron paramagnetic resonance (EPR) spectroscopy is a powerful technique to investigate chemical species with unpaired electrons. EPR spectra of CNS were recorded in the solid state and in THF solution at room temperature. As shown in Fig. 6a, CNS exhibited a typical single peak EPR signal, with a $g_e$ value of ~2.0002 in solid state and ~2.0003 in THF solution. The line widths in a magnetic field of solid state and THF solution are 0.8 and 0.4 mT, which are discordant with metal ions and comparable to the signals reported for radicaloids[57–59]. To gain more information about the structural differences between CNS and P1, we measured the EPR spectrum of P1 (Supplementary Fig. 12a). In sharp contrast, no EPR signal was observed for **P1**, indicating that the oxidative cyclodehydrogenation of **P1** to form π-conjugated carbon nanohelix structure generated a large number of magnetic entities. Metal analyses by inductively coupled plasma atomic emission spectrometry (ICP-AES) preclude any significant interference from magnetic metals (Fe, Co, Ni, etc) on the reported magnetic behavior for CNS. In addition, to clarify the light irradiation effect on the EPR signal intensity of CNS, the intensity difference of the EPR signals with and without light irradiation of CNS are shown in Fig. 6b and Supplementary Fig. 12b. After irradiating the CNS solid sample under a xenon lamp for 2 min and 3 min, no pronounced EPR signal intensity change was observed. However, when a THF solution sample was irradiated for 2 min, a strong optically induced polarization transfer was observed with the EPR signal intensity was ~2.63

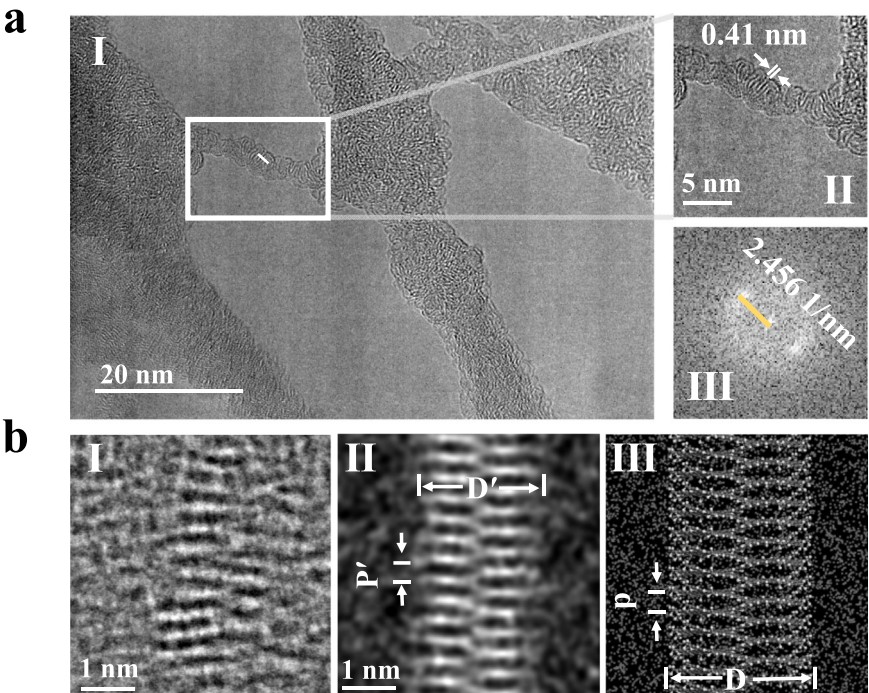

**Fig. 5 HRTEM and iDPC-STEM characterizations of CNS. a** (I) A HRTEM image showing the 2D helical self-assembled CNS. The experiment was carried out on JEM ARM-200F microscope operated at 200 kV. (II) An enlarged area HRTEM image of the white rectangle in (I). (III) The FFT pattern from the white rectangle in (I). **b** (I) Low-dose iDPC-STEM image showing the single-strand CNS. (II) Simulated projected potential (amorphous carbon substrate) of the CNS. A specific point-spread-function (PSF) width of 1.5 Å was used for CNS. Statistical analysis of CNS samples shows that measured helical pitch (**p′**) and width (**D′**) values are mainly distributed at 0.40 ± 0.03 nm and 2.7 ± 0.2 nm, respectively. (III) structural model of the CNS. Statistical analysis shows that the helical pitch (**p**) and width (**D**) calculated by DFT are ~0.41 nm and ~2.88 nm, respectively (with alkyl sidechains).

times higher than that of the unirradiated sample. One reason for this difference could be that the solid and solution states caused different samples' optical transparency, and another factor in this solvent effect might be the formation of aggregates.

The magnetization of powder samples was studied with a superconducting quantum interference device (SQUID) magnetometry. Figure 6c shows $M(H)$ data for temperatures between 1.8 and 8 K. The paramagnetic response corresponds to one $\mu_B$ of free spin in about 300 carbon atoms. The fit of a Brillouin function returns a g-factor of 2 and a $J$-value of 1.56 ± 0.13. At temperatures above 150 K, SQUID measurements indicate a Curie–Weiss 1/T decrease of the magnetization (Fig. 6d). Below 150 K, strong memory effects in the magnetization with respect to field and temperature are found. Between 150 and 50 K, we observe a small thermal hysteresis where the magnetization depends on field scan direction and field scan rate. Below 150 K, the Curie–Weiss constant initial increases before magnetic saturation at lower temperatures. Below 10 K, a large thermal hysteresis is found, where the onset of the hysteresis depends on the field scan rate. This behavior points to a complex and slow magnetic ordering behavior in CNS systems at low temperature. DFT calculation results show that the single-occupied molecular orbitals are composed of $2p$ orbital of C atoms and the whole spin density concentrates on the helix region (see Supplementary Figs. 29–30). Therefore, it is concluded that magnetic property can be attributed to the breaking of π-type electrons induced by the strain region in the helix structures. Thus, it is possible that magnetic hysteresis under 10 K is attributed to the ferromagnetic coupling between two neighboring spins in the strain region of the helix. As the temperature increases, the hysteresis becomes smaller, probably due to the effect of spin-phonon interaction.

To reveal the electrical properties of the nanostructured graphitic carbon materials, the FTO/**P1**/Au and FTO/CNS/Au devices were subjected to J-V measurements. As shown in Supplementary Fig. 13, a strong linear relationship between the current and the applied voltage (from −2 V to +2 V) was recorded, indicating an ohmic behavior of the electrical conduction. In addition, the FTO/CNS/Au device has better conductivity than the FTO/**P1**/Au, and the resistivities of the FTO/**P1**/Au and FTO/CNS/Au devices are calculated to be $1.9 \times 10^3$ and $8.4 \times 10^2$ Ω·m, respectively.

## Discussion

In summary, we report a facile bottom-up synthesis of a long-itudinally π-extended carbon nanosolenoid (CNS) from a hexaphenylbenzene precursor by a Pd-mediated Suzuki–Miyaura coupling followed by the Scholl reaction for cyclodehydrogenation. The molecular structure of CNS helix with a helical pitch of ~0.4 nm and a width of ~2.7 nm was elucidated by low-dose iDPC-STEM imaging and characterized by GPC, solid-state NMR ($^{13}$C, $^1$H MAS and 2D $^1$H-$^1$H DQ-SQ correlation spectra), FT-IR, XPS, and Raman techniques. Its photophysical properties were investigated using UV-Vis absorption, fluorescence, and TRPL spectroscopies. Notably, CNS has a low optical bandgap of 1.97 eV and intense red photoluminescence. The ground-state electronic structures and magnetic behaviors of CNS were studied by EPR, SQUID, and theoretical calculations. Magnetic testing results show that CNS has a paramagnetism response and complex magnetic ordering behavior at low temperature. Overall, such a π-conjugated CNS allows for the detailed study of its physical properties and may form the base of the development of electronic and spintronic devices containing CNS molecules.

## Methods

**Compounds preparation.** See Supplementary Information for full details regarding the synthesis and characterization of all the materials.

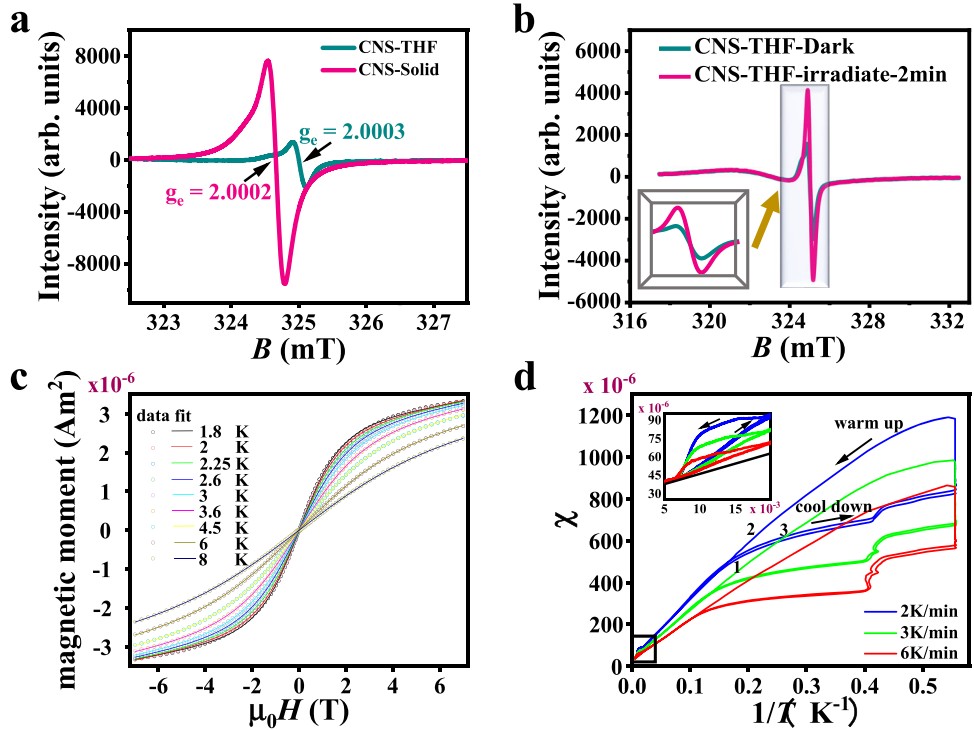

**Fig. 6 Magnetic properties of CNS. a** Room temperature EPR spectra of solid powder and THF solution of CNS. **b** EPR spectra of THF solution of CNS under dark and illumination for 2 min under 500 W xenon lamp. **c, d** Low temperature magnetization measurements of a CNS powder sample. **c** Magnetization for different temperatures (open circles, raw data) and fit of a Brillouin function with a diamagnetic background (solid lines) (field scan rates between 1 and 10 mT/s increase linearly with the applied field). **d** Sample susceptibility $\chi$ as obtained from the magnetic moment in 0.2 T applied field and an estimated sample volume of $2 \times 10^{-9}$ m$^3$ vs. Reciprocal temperature for scans from 200 K to 1.8 K cool down (1) to 200 K warm up (2) to 1.8 K cool down (3) with rates of 2 K/min (blue), 3 K/min (green), and 6 K/min (red). The kinks in the cool down above 0.4 K$^{-1}$ are due to the temperature regulation during the approach to 1.8 K and the rise in susceptibility at 1.8 K is due to 200 s relaxation intervals. The inset showed the thermal hysteresis (a magnified area of the black oblong, bottom left) between 150 and 50 K. The black line extrapolates the Curie–Weiss behavior above 150 K to lower temperatures.

**Spectroscopy analysis**. FT-IR spectra of all samples were obtained on an infrared spectrometer (Thermo-Nicolet iS10) equipped with DTGS detector and EverGlo light source using KBr disks. Chemical compositions of CNS were investigated by Thermo Scientific ESCALAB 250 X-ray photoelectron spectroscopy (XPS) equipped with Al Kα (150 W) X-ray source with a 500 μm beam spot size. The XPS measurement was performed by using monochromatic X-rays to irradiate the vacuum-dried sample under ultra-high vacuum. Raman spectrum was measured with the LabRAM HR Evolution (HORIBA Scientific) laser confocal Raman spectrometer. A 532 nm laser line was utilized with the incident power limited to as low as 0.11 mW. UV-Vis spectra were obtained on a WFZ UV-3802 scanning quasi-double beam ultraviolet-visible spectrophotometer (UNICO, Shanghai, China) equipped with environmental tritium powered lighting system (Japan) in quartz cuvettes. Steady-state fluorescence spectra were obtained with a Horiba FluoroMax-4 compact spectrofluorometer equipped with an ozone-free xenon lamp optics source. The spectra were uncorrected and quartz cells (1 cm) were used for all spectroscopic measurements. The PL lifetimes were measured by time-resolved fluorescence decay equipped with TBX picosecond photon detection module (HORIBA Scientific). ICP-AES spectrum of CNS was obtained on an IRIS Intrepid II XSP full spectrum ICP emission spectrometer (ThermoFisher, USA). EPR spectra were obtained using a JES-FA200 (JEOL, Japan) electron paramagnetic resonance spectrometer with cylindrical resonant cavity TE011 at room temperature. Conditions of EPR spectra for light irradiation samples: ES-UXL500 lighting system equipped with a 500 W UXL-500SX (Ushio Inc.) A xenon lamp was used to irradiate solid and THF solution samples of CNS.

**The iDPC-STEM imaging**. The low-dose high-resolution iDPC-STEM images were obtained under a Cs-corrected electron microscope operated at 300 kV. The beam current was reduced to 1 pA, the convergence angle was 25 mrad and the collection angle of iDPC-STEM imaging was set to 7–29 mrad. The projected electrostatic potential was simulated using the QSTEM code with a point spread function (PSF) width of 1.5 Å over a single-strand CNS structural model embedded in a 1 nm-thick amorphous carbon layer.

**Magnetic measurements**. Magnetometry data were acquired in a Quantum Design MPMS3 SQUID. The CNS sample (1.96 mg, the black powder was dried by

an exsiccator under vacuum) was filled into a polypropylene sample holder that was fixed on a DC stainless steel sample rod assembly. The diamagnetic background of the capsule was subtracted from the data. We can exclude an artifact from the machine, since a 27 mg Pt sample mounted in another polypropylene sample holder shows no hysteresis.

**Computational details**. Geometrical optimization of the whole unit cell of CNS was performed using the Vienna an initio simulation package (VASP)[60]. Perdew Burke Ernzerhof (PBE)[61] in terms of the gradient of electronic density was used to describe the exchange and correlation (XC) interaction in the Kohn–Sham equation. The interaction between ions and electrons was treated by the projector-augmented wave (PAW)[62]-based pseudopotential, which features greater computational efficiency and high accuracy. Specifically, the outer electrons of C and H were explicitly treated as valence electrons. Plane wave function with kinetic energy less than the energy of 500 eV is included in the basic set. The $1 \times 1 \times 16$ k-point grids based on the strategy of Monkhorst-Pack[63], featuring enough accuracy in the calculation of total energy via convergence test, were used to sample in the Brillouin zone. During the calculations, the convergence value was set to $1.0 \times 10^{-5}$ eV for self-consistent field calculations, and the geometrical optimization will keep been running until the <0.02 eV/Å of Hellmann–Feynman force per atom. A DFT-D3 method with Becke–Jonson damping[64] for dispersion correction was used to correct the van der Waals (vdW) interactions. Moreover, band structure and density of states were calculated by HSE-06 hybrid functional[65]. Furthermore, frontier molecular orbitals of dimers were calculated in the theoretical level of B3LYP/TZVP by using the software of ORCA.

## Data availability

Materials and methods, experimental procedures, useful information, characterization studies, [1]H NMR spectra, [13]C NMR spectra, and mass spectrometry data are available in the Supplementary Information. Additional data that support the findings of this study are available from the corresponding author upon request.

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

## Acknowledgements

This work was financially supported by the National Key Research and Development Program of China (2017YFA0402800 to P. D.), the National Natural Science Foundation of China (21971229 and 51772285 to P. D., U1932214 to S. Y., 21771161 and 22075250 to Y. Z.), the National Synchrotron Radiation Laboratory at USTC, and the Swiss National Science Foundation (SNF Project No. 206021_150784 to T. G.). We thank Mr. Zhi Wang for his help with iDPC-STEM tests.

## Author contributions

P. D. conceived and designed this research. J. W. and Y. W. synthesized the CNS material, conducted all characterizations, and EPR measurements. Y. Z. and G. S. acquired low-dose high-resolution iDPC-STEM images. G. Z. did all the calculation studies. J. W. carried out photophysical studies. Y. W. did FT-IR and $^{13}$C NMR measurements. T. G. performed and analyzed SQUID experiments. J. W., Y. Z., G. Z., Y. W., S. W., P. H., G. S., M. C., S. Y., T. G., and P. D. co-wrote the paper, and all the authors commented on it.

## Competing interests

The authors declare no competing interests.
