## [Peer Review File · Nature Communications]

Reviewers' Comments:

Reviewer #1:

Remarks to the Author:

The authors have synthesized, bottom-up, carbon nano-solenoids (CNS) and conducted extensive characterization via structural (Raman and microscopy), optical (photoluminescence), and magnetic measurements. While I find the synthesis of the CNS to be impressive, some synthesis of helical-nanosolenoid structures family has been done (Refs 26-27 here, and T. H. Ly et al *Advanced Materials*, 28, 7723 (2016)). Still, new synthesis is desirable and welcome.

Unfortunately, the characterization of the CNS appears extensive yet a bit superficial, offering little insight into the physical properties of the CNS. Specifically:

1) Paper seems somehow unfinished. For example, about half of Figure 1 is not discussed in the text at all. In the caption they mention structure "after twist", but twist never mentioned in the text (what twist? Do they simply mean that the view is tilted? Not Eshelby twist?). Figure 1d (II, III and IV) are clearly in a perspective view that makes exact reading the image, e. g. of the stacking, pitch, impossible. The (d)-I should cite in caption again [25] which shows same structures, even on the Nano Lett cover <https://pubs.acs.org/toc/nalefd/16/1>

2) Precise structural control of the final material seems to be the major claim of this paper. But no atomistic structural characterization was done. What is the structure of the edges (paper hints it is zigzag)? What is the structure of the helical core, near its axis? One should be able to determine the stacking, if the "planes-loops" are epitaxial above each other, or twisted? (Can all of those parameters be changed with the use of different precursors?)

3) Much of the characterization seems to be extremely tech-detail oriented, yet offering little interpretation and insight into the properties of the CNS itself. For instance, the "Photophysical Property" section merely reports a single bandgap 1.97 eV, while existing theories suggest that the bandgap of CNS sensitively depends on its structure, width, edge, etc. Does 1.97 eV correspond to the size of the CNS?

4) Inspired by theory in ref. 25, the authors measure magnetism of the CNS; however there seems to be disconnect. While the focus of ref. 25 is how a current through the high-pitch CNS creates large magnetic field, but the measurements in this work is on the intrinsic magnetism of the CNS, no current. Geometrical/Riemann inspiration is valid, but magnetism is different. Could they try applying voltage to the solenoid (like paper above by T. Ly et al *Adv Mater*)? that would reveal real solenoid behavior.

Here the magnetism comes from externally applied magnetic field. They found small hysteresis in magnetization at T=50-150K, and large hysteresis under 10K, why? What is the cause of hysteresis? Is it a strange property of individual molecules per se or their collective behavior in a sample?

To me, if the synthesis is successful, the key appeal would be in Fig. 5, the direct structural evidence, but with much improved quality: HRTEM or any methods which would show structures there with state-of-the art resolution. Second desirable is the definitive solenoid test, or at least current demonstration as in Ly et al. *Adv Mater*.

Reviewer #2:

Remarks to the Author:

Dear Editor

The work of Wang and coworkers reports on the synthesis of cove-edge/armchair-edge type graphene nanosolenoids via Pd-mediated Suzuki-Miyaura coupling of different phenylene precursors and subsequent Scholl dehydrogenation.

The fabrication of graphene nanosolenoids is a significant milestone in the field of carbon allotropes and set to spark interest in their unexplored physical properties.

The claim of new graphene nanoribbons, let alone, nanosolenoids, usually mandates unambiguous structural evidence such as MALDI-ToF, Raman-specific modes or synthesis of model dimer or n-mer compounds, which are unfortunately missing in the current work.

Research towards the demonstration of graphene nanosolenoids is a highly-competitive field. Thus non-standard structure-property characterizations should, in principle, be a valid point of departure to substantiate the authors' claims. While AFM and STM characterization could provide the missing structural information, the authors do not unambiguously measure the nanosolenoids "diameter" from the height profile of the material (i.e. the AFM height when adsorbing edge-on) which entails a high degree of confidence (a 3D molecular mechanics model would help with the actual dimensions). For ambient conditions, the AFM data can be considered of reasonable quality if the authors specify additional information such as tips employed, AFM parameters, the exact deposition parameters, substrate treatments and include large-scale ($\sim 10000 \text{ nm}^2$) image surveys. The authors should further consider that H-Si(100)2x1 and possibly also O-Si(100)2x1 substrates feature parallel atomic dimer rows which could mimic graphene nanoribbons or nanosolenoids. The authors measure an extraordinarily large pitch of $b \sim 2.5 \text{ nm}$, differing from the graphite interlayer distance by approx. a factor of 10. It's difficult to rationalize this by t-butyl units alone. Careful molecular modelling is also critical here, especially since the lead angle of such Riemann helicoid amounts possibly to $\arctan(b/2\pi r) \sim 10^\circ$, which is lower than 45° as the authors claim.

Finally, magnetism appears to be an unexpected property of cove/armchair -type nanosolenoids: In ref. 25, magnetic nanosolenoids appear to be zigzag-edge -type nanoribbons and not cove/armchair type. Thus, quantum chemistry analysis is possibly required to rationalize and explore the open-shell character of cove-type nanosolenoids vis-à-vis cove-type nanoribbons.

A large lead angle and magnetism might be speculatively explained by intercalated DDQ molecules in the polymer P1 or in the nanosolenoid—an important discovery by itself. Here again, MALDI is an essential tool to rationalize such impurities.

In summary, the authors are encouraged to resubmit at a later stage, carefully addressing the 3D structural-magnetic property relationship of their system.

Point-by-point response to reviewer's comments:

Reviewer #1

The authors have synthesized, bottom-up, carbon nano-solenoids (CNS) and conducted extensive characterization *via* structural (Raman and microscopy), optical (photoluminescence), and magnetic measurements. While I find the synthesis of the CNS to be impressive, some synthesis of helical-nanosolenoid structures family has been done (Refs 26-27 here, and T. H. Ly et al *Advanced Materials*, 28, 7723 (2016)). Still, new synthesis is desirable and welcome. Unfortunately, the characterization of the CNS appears extensive yet a bit superficial, offering little insight into the physical properties of the CNS.

Response: We thank this reviewer very much for these good comments and suggestions to help us improve our manuscript. As suggested, we have provided more characterization data of CNS using the combination of FT-IR, Raman, solid-state ^1H MAS NMR, solid-state ^{13}C NMR, solid-state 2D ^1H - ^1H DQ-SQ correlation NMR, XRD, SEM, AFM, HR-TEM, J - V measurement of FTO/CNS/Au device, optical profilometer and theoretical calculations in the revised manuscript.

Specifically:

1) Paper seems somehow unfinished. For example, about half of Figure 1 is not discussed in the text at all. In the caption they mention structure "after twist", but twist never mentioned in the text (what twist? Do they simply mean that the view is tilted? Not Eshelby twist?).

Response: We thank the reviewer for pointing out this problem. We initially wanted to express that the view is tilted. To avoid confusion, we have replaced Figure 1d-II in our previous manuscript and removed "after twist" and corrected the corresponding parts (*Caption to Figure 1d*) as follows:

Caption to Figure 1d: "...(d) (I) Design of the novel 3D fully π -extended curved single-stranded carbon nanosolenoids (CNS).²⁵ (II-IV) The atomic structure of CNS with side view (II), diagrammatic sketch (III) and axial view (IV) by the DFT calculations. (V) Edges and helical core structures of CNS..."

Figure 1d. (d) (I) Design of the novel 3D fully π -extended curved single-stranded carbon nanosolenoids (CNS).²⁵ (II-IV) The atomic structure of CNS with side view (II), diagrammatic sketch (III) and axial view (IV) by the DFT calculations. (V) Edges and helical core structures of CNS.

Figure 1d (II, III and IV) are clearly in a perspective view that makes exact reading the image, e. g. of the stacking, pitch, impossible. The (d)-I should cite in caption again [25] which shows same structures, even on the *Nano Lett* cover <https://pubs.acs.org/toc/nalefd/16/1>

Response: We thank the reviewer for this constructive suggestion. As suggested, we provided the atomic structure analysis of the CNS (*page 6, in the first paragraph*) and cited the reference [25] (*Nano Lett.* **2016**, 16, 34-39) in the *Caption to Figure 1d*. We have replaced Figure 1d (II, III and IV) in our initial manuscript and corrected the caption of *Figure 1d*. And the related description (*page 6, in the first paragraph*) has also been revised as follows:

Caption to Figure 1d: “...(d) (I) Design of the novel 3D fully π -extended curved single-stranded carbon nanosolenoids (CNS).²⁵ (II-IV) The atomic structure of CNS with side view (II), diagrammatic sketch (III) and axial view (IV) by the DFT calculations. (V) Edges and helical core structures of CNS...”

“...To investigate atomic structure information such as width, edges, helical core, stacking, band, spin density, molecular orbitals and so on, density-functional theory (DFT) calculations were carried out for the geometric structure and electronic structure of CNS using the Vienna an initio simulation package (VASP). The results are summarized in Figure 1d (II, III and IV), Figure S29, Figure S30 and Table S1. Figure 1d-II shows the geometrical optimization structure of CNS, and the results indicate that CNS features helix-bundles configuration along the Burgers vector **b** (parallel to the *c*-axis marked in yellow, see Figure 1d-III) with the d-spacing of 4.118 Å. Furthermore, molecular dynamic simulation results at 300 K also reveal the thermal stability of the bundles configuration (see the attached movie). Figure 1d-IV shows an axial view of CNS. CNS is a novel carbon material that has armchair edges and armchair helical core structures (marked in Figure 1d-V)...”

2) Precise structural control of the final material seems to be the major claim of this paper. But no atomistic structural characterization was done. What is the structure of the edges (paper hints it is zigzag)? What is the structure of the helical core, near its axis? One should be able to determine the stacking, if the “planes-loops” are epitaxial above each other, or twisted? (Can all of those parameters be changed with the use of different precursors?)

Response: We thank the reviewer for pointing out this problem. As suggested, we have provided the atomic structure analysis of the CNS (*page 6, in the first paragraph*) in our revised manuscript. CNS is a novel carbon material that possesses armchair edges and armchair helical core structures, and the edge and helical core structures have been marked in Figure 1d-V in the revised manuscript. Moreover, the structure of precursors determines the type of edges of the target compound, which has been reported in

previous studies of small curved graphene nanoribbons (see ref: *Chem. Sci.* **8**, 1068-1074 (2017); *Angew. Chem. Int. Ed.* **57**, 14782-14786 (2018); *Chem. Sci.* **9**, 3917-3924 (2018); *Angew. Chem. Int. Ed.* **58**, 8068-8072 (2019); *Angew. Chem. Int. Ed.* **59**, 7139-7145 (2020)). The related description (*page 6, in the first paragraph*) has been revised as follows:

“...To investigate atomic structure information such as width, edges, helical core, stacking, band, spin density, molecular orbitals and so on, density-functional theory (DFT) calculations were carried out for the geometric structure and electronic structure of CNS using the Vienna an initio simulation package (VASP). The results are summarized in Figure 1d (II, III and IV), Figure S29, Figure S30 and Table S1. Figure 1d-II shows the geometrical optimization structure of CNS, and the results indicate that CNS features helix-bundles configuration along the Burgers vector **b** (parallel to the *c*-axis marked in yellow, see Figure 1d-III) with the d-spacing of 4.118 Å. Furthermore, molecular dynamic simulation results at 300 K also reveal the thermal stability of the bundles configuration (see the attached movie). Figure 1d-IV shows an axial view of CNS. CNS is a novel carbon material that has armchair edges and armchair helical core structures (marked in Figure 1d-V)...”

Figure 1d. (d) (I) Design of the novel 3D fully π -extended curved single-stranded carbon nanosolenoids (CNS).²⁵ (II-IV) The atomic structure of CNS with side view (II), diagrammatic sketch (III) and axial view (IV) by the DFT calculations. (V) Edges and helical core structures of CNS.

The stacking of CNS can be analyzed by ^1H NMR magic-angle spinning (MAS) spectrum, 2D ^1H - ^1H double-quantum single-quantum (DQ-SQ) NMR correlation spectrum combined with the theoretical calculations. The related descriptions (see *pages S16-S17 in our revised Supplementary Information, pages 8-9 in our revised manuscript*) have been revised as follows:

“...Aromatic protons were significantly reduced and the ^1H line width significantly increased by comparing the ^1H NMR magic-angle spinning (MAS) spectra of **P1** and CNS, indicating that **P1** with a semi-flexible to semi-rigid structure becomes rigid after graphitization to CNS (Figure S6a). Moreover, by comparing the solid-state 2D ^1H - ^1H double-quantum single-quantum (DQ-SQ) NMR correlation spectra of **P1** and CNS

(see Figures S6b-6c), the results show that the ^1H - ^1H auto-correlation signals of the aromatic protons in **P1** disappeared after graphitization to **CNS** and the aromatic protons in **CNS** are far away from each other, which is consistent with the structural characteristics of **CNS**. The ^1H NMR signals of bulk **CNS** are significantly broadened, possibly due to the heterogeneous packing of **CNS**, and the currents of aromatic/anti-aromatic rings cause the ^1H NMR signals to shift in the opposite direction.⁸...”

“...Figure S6 shows the ^1H MAS NMR and 2D ^1H - ^1H DQ-SQ NMR spectra for **P1** and **CNS**. The comparison of the ^1H MAS NMR spectra (Figure S6a) of **P1** and **CNS** demonstrates an obvious increase in ^1H line width after graphitization. Figures S6b-6c show the solid-state 2D ^1H - ^1H DQ-SQ NMR correlation spectra of **P1** and **CNS**. The spectrum of **P1** in Figure S6b has the expected ^1H - ^1H auto-correlation signals (SQ = 0.7/DQ = 1.4 ppm and SQ = 6.6/DQ = 13.2 ppm). The ^1H - ^1H cross-correlation signals can be generated intramolecularly since the *tert*-butyl groups are connected to a phenyl group. For **CNS**, the ^1H - ^1H auto-correlation signals of the aliphatic protons can be straightforwardly assigned (SQ = 2.0/DQ = 4.0 ppm). There are almost no ^1H - ^1H auto-correlation signals of the aromatic protons, which is consistent with the feature that the aromatic protons in **CNS** are far away from each other. The ^1H - ^1H cross-correlation between the *tert*-butyl groups and the aromatic protons (SQ = 9.7/DQ = 11.7 ppm) can also be result from intramolecular interactions....”

Figure S6. (a) ^1H MAS NMR spectra of **P1** (top) and **CNS** (bottom). (b) The 2D ^1H - ^1H DQ-SQ correlation spectra shown for **P1** (b) and **CNS** (c) were recorded employing 4 rotor periods DQ excitation using the Back-to-Back (BaBa) DQ recoupling scheme with an XY16 phase cycle.

All of these parameters could be tuned by using different precursors, such as larger steric groups or small functional groups. Repulsion or attraction between different groups leads to a wider or narrower π - π stacking distance, which has been reported in previous studies (see ref: *Nat. Chem.* **6**, 126-132 (2014); *J. Am. Chem. Soc.* **142**, 18293-18298 (2020)). Recently, our laboratory also synthesized a CNS derivative polymer (unpublished results) containing N atoms using different precursors, which also proves that all of those parameters can be changed by using different precursors.

- 3) Much of the characterization seems to be extremely tech-detail oriented, yet offering little interpretation and insight into the properties of the CNS itself. For instance, the "Photophysical Property" section merely reports a single bandgap 1.97 eV, while existing theories suggest that the bandgap of CNS sensitively depends on its structure, width, edge, etc. Does 1.97 eV correspond to the size of the CNS?

Response: We thank the reviewer for pointing out this. It is known that the bandgaps of graphene nanoribbons are sensitive to the sizes, structures, widths, edges, etc (*Nat. Chem.* **4**, 699-704 (2012); *Nature*, **531**, 489-493 (2016)). Therefore, it is possible that the CNS is sensitive to the sizes. Because CNS is a polymer and we can not separate different sizes. 1.97 eV should correspond to the whole material in a diluted solution. It is also sensitive to the edges. Recently, our laboratory synthesized a CNS derivative polymer containing N atom edges with an optical bandgap of 1.90 eV (see following Figure A, **unpublished results**), which also confirmed that the bandgap of CNS sensitively depends on its structure, width, edge, etc. As suggested, we have revised the interpretation and insight into the properties of the CNS itself in our revised manuscript. The related descriptions (*pages 10 and 15*) have been revised as follows:

“...The bandgap of GNRs sensitively depends on its structure, width, edge, etc. and the absorption onset wavelength (λ_{onset}) of CNS was observed at ~ 629 nm, corresponding to the optical bandgap (E_{gap}) is ~ 1.97 eV.⁸...”

“...DFT calculation results show that the single-occupied molecular orbitals are composed of $2p$ orbital of C atoms and the whole spin density concentrates on the helix region (see Figures S29-S30). Therefore, it is concluded that magnetic property can be attributed to the breaking of π -type electrons induced by the strain region in the helix structures. Thus, it is possible that magnetic hysteresis under 10 K is attributed to the ferromagnetic coupling between two neighboring spins in the strain region of the helix. As the temperature increases, the hysteresis becomes smaller, probably due to the effect of spin-phonon interaction...”

Figure A. UV-Vis absorption spectrum (left) for a CNS derivative polymer containing N atoms (N-CNS, right) in DMF. The absorption onset wavelength (λ_{onset}) of the N-CNS was observed at ~ 652 nm, corresponding to the optical bandgap (E_{gap}) is ~ 1.90 eV.

- 4) Inspired by theory in ref. 25, the authors measure magnetism of the CNS; however there seems to be disconnect. While the focus of ref. 25 is how a current through the high-pitch CNS creates large magnetic field, but the measurements in this work is on the intrinsic magnetism of the CNS, no current. Geometrical/Riemann inspiration is valid, but magnetism is different. Could they try applying voltage to the solenoid (like paper above by T. Ly et al Adv Mater)? that would reveal real solenoid behavior.

Response: We thank the reviewer for pointing out this. The large magnetic field and superior inductance properties of this spiral tubular nanocarbon material are very attractive when applied voltage and the electrical currents flow helically. We are also inspired by the work [25] (*Nano Lett.* **2016**, 16, 34-39) and very much eager to do so. However, this measurement is very difficult for us since there is no such an equipment available in our school or even around us. But fortunately, we found that CNS possesses an inherently magnetic property, which is also an interesting feature. So we decided to introduce our work to the science society using the intrinsic magnetism of the CNS as a highlight. In addition, as suggested, the J - V measurements for the FTO/P1/Au and FTO/CNS/Au devices were conducted, as shown in Figure S13 (see Supplementary Information). The related description (*page 15, in the second paragraph*) has been revised as follows:

“...To reveal the electrical properties of the nanostructured graphitic carbon materials, the FTO/P1/Au and FTO/CNS/Au devices were subjected to J - V measurements. As shown in Figure S13, the strong linear relationship between the current and the applied voltage (from -2V to +2V) was recorded, indicating an ohmic behavior of the electrical conduction. In addition, the FTO/CNS/Au device has better conductivity than the FTO/P1/Au, and the resistivities of the FTO/P1/Au and FTO/CNS/Au devices are calculated to be 1.9×10^3 and $8.4 \times 10^2 \Omega \cdot \text{m}$, respectively...”

Figure S13. J - V profiles of a cast film of **P1** and **CNS**. Inset: The FTO/**P1**/Au and FTO/**CNS**/Au devices for an electrical conductivity tests. The samples thickness are ~ 161 nm for **P1** and ~ 265 nm for the **CNS** when the average value is taken after three tests. (b) Schematic diagram of the FTO/**P1**/Au or FTO/**CNS**/Au device.

Figure B. Test the sample thickness on the FTO/**P1**/Au and FTO/**CNS**/Au devices using a Profiler (KLA-Tencor, USA).

Here the magnetism comes from externally applied magnetic field. They found small hysteresis in magnetization at $T=50$ - 150 K, and large hysteresis under 10 K, why? What is the cause of hysteresis? Is it a strange property of individual molecules per se or their collective behavior in a sample?

Response: We thank the reviewer for this point. As the referee mentions, the magnetisation measurements presented in Figures 6 (c) and (d) rely on an applied external magnetic field. Figure 6c shows, in line with the EPR results of Figures 6a-6b that the **CNS** has a relatively high density of unpaired spins. Experimentally, we have completely ruled out the effect to be due to paramagnetic metal ions.

The deviation from a Curie Weiss $1/T$ susceptibility shows that the unpaired spins in the sample interact, and undergo ordering. The two different thermal hystereses furthermore indicate that the magnetic ordering is complex. DFT calculations suggest that the single-occupied molecular orbitals are composed of $2p$ orbitals of C atoms and the whole spin density concentrates on the helix region (see Figures S29-S30). Therefore, we propose that the paramagnetic properties are due to the breaking of π -

type electrons in the strain region in the helix region. At this stage of knowledge an answer on "why" this peculiar electronic structure results in two different orderings that are reflected in the magnetic susceptibility below 150 K has to recall concepts like freezing transitions.

Likely, the hystereses are related to structural transitions in the sample like it is e.g. observed in pentacene films (*Appl. Phys. Lett.* **99**, 211102 (2011); <https://doi.org/10.1063/1.3663863>). For the CNS this could be a change in the twist of the carbon layers and a reorientation of the molecules. The fact that we observe this transition in the magnetic susceptibility demonstrates on how sensitive magnetic susceptibility measurements can be if there are free spins, and that this could give rise to new spintronic applications. At this stage we may not answer the question on whether both hystereses are single molecule or crystal properties.

The twist of carbon materials can induce magnetism, which has been recently reported in the literature. Such as in 2010, N. Levy et al reported pseudo-magnetic fields greater than 300 Tesla induced by strain in graphene nanobubbles (see ref: *Science* **321**, 385-388 (2010)). Yuliang Li and coworkers reported a stable twisted 2D hydrocarbon tetraradical in 2016 (see ref: *Nat. Commun.* **7**, 1-11 (2016)), indicating that distortions between atomic layers in carbon material can generate electrons out of the domain or free electrons, making them magnetic. For a planar molecule or a single molecule, it is a strange property. But for the twisted molecule, it has been reported that the twist in carbon materials can also generate free electrons and hence magnetism (see ref: *Nat. Commun.* **7**, 1-11 (2016)).

Figure S29. Band structure with high-symmetry points (G: 0.0, 0.0, 0.0; Z: 0.0, 0.0, 0.5; F: 0.0, 0.5, 0.0; Q: 0.0, 0.5, 0.5) and density of states of CNS.

Figure S30. Spin density (a) and single-occupied molecular orbitals (b-c) of CNS.

To me, if the synthesis is successful, the key appeal would be in Fig. 5, the direct structural evidence, but with much improved quality: HRTEM or any methods which would show structures there with state-of-the-art resolution. Second desirable is the definitive solenoid test, or at least current demonstration as in Ly et al. Adv Mater.

Response: We agree with the reviewer's good suggestion! Following the reviewer's suggestion, the detailed structural characterizations of the CNS were further investigated by XRD, HRTEM combined with theoretical calculations. The related description (*pages 12-13*) has been revised as follows:

“...Powder XRD diffraction shows that CNS has good crystallinity (Figure S8). Then, the molecular structure of CNS with such high molecular weights can be visualized at the solid-air interface using HRTEM technique. Figure 5c displays a large-scale TEM image of a dry film of CNS. Interestingly, the TEM image reveals helical and disordered CNS self-assembled into domains with characteristic spring-like structures (Figure 5c-I and Figure S9), which probably result from the helical structure characteristics and magnetic properties of CNS. The enlarged image of the area is shown in Figure 5c-II. The measured width of the stripes is in accord with the calculation helical interlamellar spacing of CNS (ca. 4.118 Å). Figure 5c-III is the Fast Fourier Transfer (FFT) of the white rectangle in Figure 5c-I. Furthermore, the HRTEM images in Figure S10 show clear lattice fringes corresponding to XRD patterns of self-assembly CNS, and the lattice fringe spacings are 2.65, 1.92 and 1.11 Å, respectively, which could be due to the self-assembly of CNS polymer dimers and trimers. Moreover, the wide-field SEM images also show the self-assembly of CNS (Figure S11)...”

Figure S8. XRD patterns of CNS.

Figure 5c. (c) (I) A HRTEM image showing the 2D helical self-assembled CNS. The experiment was carried out on JEM ARM-200F microscope operated at 200 kV. (II) An enlarged area HRTEM image of the white rectangle in (I). (III) The Fast Fourier Transfer from the white rectangle in (I).

Figure S9. HRTEM images with different fields of view and different resolutions of CNS. The experiment was carried out on JEM ARM-200F microscope.

(a)

Figure S10. HRTEM images with different fields of view and different resolutions of CNS. The experiment was carried out on JEM-2100F microscope.

Figure S11. SEM images of self-assembled CNS at different resolutions and views.

The large magnetic field and superior inductance properties of this spiral tubular nanocarbon material are very attractive when applied voltage and the electrical currents flow helically. We are also inspired by the work [25] (*Nano Lett.* **2016**, 16, 34-39) and very much eager to do so. However, this measurement is very difficult for us since there is no such equipment available in our school or even around us. But fortunately, we found that CNS possesses an inherently magnetic property, which is also an interesting feature. So we decided to introduce our work to the science society using the intrinsic

magnetism of the **CNS** as a highlight. In addition, as suggested, the J - V measurements for the FTO/**P1**/Au and FTO/**CNS**/Au devices were conducted, as shown in Figure S13 (see Supplementary Information). The related description (*page 15, in the second paragraph*) has been revised as follows:

“...To reveal the electrical properties of the nanostructured graphitic carbon materials, the FTO/**P1**/Au and FTO/**CNS**/Au devices were subjected to J - V measurements. As shown in Figure S13, the strong linear relationship between the current and the applied voltage (from -2V to +2V) was recorded, indicating an ohmic behavior of the electrical conduction. In addition, the FTO/**CNS**/Au device has better conductivity than the FTO/**P1**/Au, and the resistivities of the FTO/**P1**/Au and FTO/**CNS**/Au devices are calculated to be 1.9×10^3 and $8.4 \times 10^2 \Omega \cdot \text{m}$, respectively...”

Figure S13. J - V profiles of a cast film of **P1** and **CNS**. Inset: The FTO/**P1**/Au and FTO/**CNS**/Au devices for an electrical conductivity tests. The samples thickness are ~ 161 nm for **P1** and ~ 265 nm for the **CNS** when the average value is taken after three tests. (b) Schematic diagram of the FTO/**P1**/Au or FTO/**CNS**/Au device.

Overall, we thank this reviewer very much for these helpful comments and suggestions to help us improve our manuscript.

Reviewer #2

The work of Wang and coworkers reports on the synthesis of cove-edge/armchair-edge type graphene nanosolenoids *via* Pd-mediated Suzuki-Miyaura coupling of different phenylene precursors and subsequent Scholl dehydrogenation. The fabrication of graphene nanosolenoids is a significant milestone in the field of carbon allotropes and set to spark interest in their unexplored physical properties.

Response: We thank this reviewer very much for these helpful comments to help us improve the manuscript.

The claim of new graphene nanoribbons, let alone, nanosolenoids, usually mandates unambiguous structural evidence such as MALDI-TOF, Raman-specific modes or synthesis of model dimer or *n*-mer compounds, which are unfortunately missing in the current work.

Response: We thank this reviewer very much for these helpful comments to help us improve the manuscript. In fact, we tried to use different matrix (trans-2-[3-(4-tert-butylphenyl)-2-methyl-2-propenylidene]malononitrile (DCTB) or 7,7,8,8-tetracyanoquinodimethane (TCNQ)) with/without AgCOCF₃ as an ionization additive, and a standard MALDI sample preparation method was used: a droplet of 1.5 μL of 1:1 *v/v* matrix/analyte solution was deposited on the target plate to cover the target surface completely and allowed to dry in 20-30 min. However, our dry samples can not be absorbed on the plate. Thus, it is very difficult for the samples and matrix to form a co-crystalline film, and only very few very small-samples spots can be seen in the field of view. By changing different test parameters, including mass range selector, laser, automation, digitizer, laser attenuator, laser focus, detector gain voltage offset, electronic gain button definitions and spectrometer parameters such as high voltage, polarity, pulsed ion beam, matrix suppression mode, etc., and then irradiated these small samples spots with a laser, the mass data of **P1** shown in the following Figure A. The characteristic peak patterns of the polymer were obtained and many **P1** polymer fragments with different molecular masses can be seen, but considering the limitation of MALDI-TOF Mass analysis for high-molecular-weight polymers with a broad molecular weight distribution (see ref: *Angew. Chem. Int. Ed.* **50**, 2540-2543 (2011); *J. Am. Chem. Soc.* **134**, 18169-18172 (2012); *Nat. Chem.* **6**, 126-132 (2014)), the highest mass weight detected in this measurement did not correspond to the largest polymer in the sample. The Raman spectroscopy (Figure 3e) in our initial manuscript has shown a distinct fine structure with strong G and D bands, typical of the nanostructures of GNR (see ref: *J. Am. Chem. Soc.* **134**, 18169-18172 (2012); *Nature* **466**, 470-473 (2010)). Although the model dimer or *n*-mer compounds can be used as a suitable reference compound to help analyze the structure of the CNS, it is regrettable that we can not synthesize them within the limited time. Although the model dimer or *n*-mer compounds were not available, all these characterization data from NMR (¹H MAS NMR, solid-state ¹³C NMR, 2D ¹H-¹H DQ-SQ NMR), Gel permeation chromatography,

FT-IR spectrum, XPS spectrum, Raman spectrum, fluorescence spectrum, AFM images, HRTEM images and theoretical calculations can provide evidences to confirm that the CNS had been successfully prepared.

Figure A. HR-MS (MALDI-TOF) data for P1.

Research towards the demonstration of graphene nanosolenoids is a highly-competitive field. Thus non-standard structure-property characterizations should, in principle, be a valid point of departure to substantiate the authors' claims. While AFM and STM characterization could provide the missing structural information, the authors do not unambiguously measure the nanosolenoids "diameter" from the height profile of the material (i.e the AFM height when adsorbing edge-on) which entails a high degree of confidence (a 3D molecular mechanics model would help with the actual dimensions). For ambient conditions, the AFM data can be considered of reasonable quality if the authors specify additional information such as tips employed, AFM parameters, the exact deposition parameters, substrate treatments and include large-scale ($\sim 10000 \text{ nm}^2$) image surveys. The authors should further consider that H-Si(100)2x1 and possibly also O-Si(100)2x1 substrates feature parallel atomic dimer rows which could mimic graphene nanoribbons or nanosolenoids. The authors measure an extraordinarily large pitch of $b \sim 2.5 \text{ nm}$, differing from the graphite interlayer distance by approx. a factor of 10. It's difficult to rationalize this by tertbutyl units alone. Careful molecular modelling is also critical here, especially since the lead angle of such Riemann helicoid amounts possibly to $\arctan(b/2\pi r) \sim 10^\circ$, which is lower than 45° as the authors claim.

Response: We thank this reviewer very much for these helpful comments to help us improve the manuscript. Following the reviewer's suggestion, the 3D molecular mechanics model (see Figure 1d in our revised manuscript and Figures S29-S30 in our revised Supplementary Information) and detailed AFM testing information have been provided in our revised manuscript. The related description (page S18 in our revised

Supplementary Information and page 6 in our revised manuscript) has been revised as follows:

“...To investigate atomic structure information such as width, edges, helical core, stacking, band, spin density, molecular orbitals and so on, density-functional theory (DFT) calculations were carried out for the geometric structure and electronic structure of CNS using the Vienna *ab initio* simulation package (VASP). The results are summarized in Figure 1d (II, III and IV), Figure S29, Figure S30 and Table S1. Figure 1d-II shows the geometrical optimization structure of CNS, and the results indicate that CNS features helix-bundles configuration along the Burgers vector **b** (parallel to the *c*-axis marked in yellow, see Figure 1d-III) with the d-spacing of 4.118 Å. Furthermore, molecular dynamic simulation results at 300 K also reveal the thermal stability of the bundles configuration (see the attached movie). Figure 1d-IV shows an axial view of CNS. CNS is a novel carbon material that has armchair edges and armchair helical core structures (marked in Figure 1d-V)...”

“...The stock solution of CNS in dry THF (0.02 mg/mL) was prepared. Samples for the AFM measurements of CNS were prepared by dropping aliquots (ca. 40 µL) of the stock solutions on a freshly cleaved (100) p-type silicon wafer doped with boron at room temperature (ca. 25 °C), and the solvents were slowly evaporated under a benzene vapor atmosphere. The benzene vapor was prepared by putting 1 mL of benzene into a 2 mL bottle, and then placed the 2 mL bottle and the silicon wafer substrates simultaneously in a 50 mL bottle. The silicon wafer substrates were then dried under vacuum for 2 hours before AFM measurements...”

We agree with the reviewer that H-Si(100)2x1 and possibly also O-Si(100)2x1 substrates feature parallel atomic dimer rows which could mimic graphene nanoribbons or nanosolenoids. However, the AFM image in our present study was obtained in a (100) p-type silicon wafer doped with boron, and we did not use an H-Si(100)2x1 and O-Si(100)2x1 substrates. To further exclude this possibility, we also tested AFM image using a blank (100) p-type silicon wafer. The (100) p-type silicon wafer does not show parallel atomic dimer rows, confirming that the AFM images in our study are the morphologies from our sample.

Furthermore, the detailed structural characterizations of CNS were further investigated by XRD, HRTEM combined with theoretical calculations. The related description (*pages 12-13*) has been revised as follows:

“...Powder XRD diffraction shows that CNS has good crystallinity (Figure S8). Then, the molecular structure of CNS with such high molecular weights can be visualized at the solid-air interface using HRTEM technique. Figure 5c displays a large-scale TEM image of a dry film of CNS. Interestingly, the TEM image reveals helical and disordered CNS self-assembled into domains with characteristic spring-like structures (Figure 5c-I and Figure S9), which probably result from the helical structure characteristics and

magnetic properties of CNS. The enlarged image of the area is shown in Figure 5c-II. The measured width of the stripes is in accord with the calculation helical interlamellar spacing of CNS (ca. 4.118 Å). Figure 5c-III is the Fast Fourier Transfer (FFT) of the white rectangle in Figure 5c-I. Furthermore, the HRTEM images in Figure S10 show clear lattice fringes corresponding to XRD patterns of self-assembly CNS, and the lattice fringe spacings are 2.65, 1.92 and 1.11 Å, respectively, which could be due to the self-assembly of CNS polymer dimers and trimers. Moreover, the wide-field SEM images also show the self-assembly of CNS (Figure S11)....”

Figure S8. XRD patterns of CNS.

Figure 5c. (c) (I) A HRTEM image showing the 2D helical self-assembled CNS. The experiment was carried out on JEM ARM-200F microscope. (II) An enlarged area HRTEM image of the white rectangle in (I). (III) The Fast Fourier Transfer from the white rectangle in (I).

Figure S9. HRTEM images with different fields of view and different resolutions of CNS. The experiment was carried out on JEM ARM-200F microscope.

Figure S10. HRTEM images with different fields of view and different resolutions of CNS. The experiment was carried out on JEM-2100F microscope.

Figure S11. SEM images of self-assembled CNS at different resolutions and views.

Finally, magnetism appears to be an unexpected property of cove/armchair -type nanosolenoids: In ref. 25, magnetic nanosolenoids appear to be zigzag-edge -type nanoribbons and not cove/armchair type. Thus, quantum chemistry analysis is possibly required to rationalize and explore the open-shell character of cove-type nanosolenoids vis-à-vis cove-type nanoribbons.

Response: We agree with the reviewer's good suggestion! Following the reviewer's

suggestion, pin-polarized density functional theory calculations were performed in our revised manuscript. Computational results revealed that the magnetic property of CNS can be due to the strained carbon structures (see Figures S29-S30 in our revised Supplementary Information). The related description (page15, in the first paragraph) has been revised as follows:

“...DFT calculation results show that the single-occupied molecular orbitals are composed of $2p$ orbital of C atoms and the whole spin density concentrates on the helix region (see Figures S29-S30). Therefore, it is concluded that magnetic property can be attributed to the breaking of π -type electrons induced by the strain region in the helix structures. Thus, it is possible that magnetic hysteresis under 10 K is attributed to the ferromagnetic coupling between two neighboring spins in the strain region of the helix. As the temperature increases, the hysteresis becomes smaller, probably due to the effect of spin-phonon interaction...”

Figure S29. Band structure with high-symmetry points (G: 0.0, 0.0, 0.0; Z: 0.0, 0.0, 0.5; F: 0.0, 0.5, 0.0; Q: 0.0, 0.5, 0.5) and density of states of CNS.

Figure S30. Spin density (a) and single-occupied molecular orbitals (b-c) of CNS.

A large lead angle and magnetism might be speculatively explained by intercalated DDQ molecules in the polymer P1 or in the nanosolenoid-an important discovery by itself. Here again, MALDI is an essential tool to rationalize such impurities.

Response: We thank the reviewer for pointing out this. However, the CNS sample that was conducted for the magnetic test was purified by column chromatography and Soxhlet extraction. High-resolution mass spectrometry (HR-MS) has also verified that the sample did not contain any DDQ molecules (see following Figure B). In order to explore the reason of the magnetic behavior of CNS, the CNS sample was prepared by using FeCl₃ oxidative cyclodehydrogenation of P1. The same methods were used to purified the CNS sample and inductively coupled plasma atomic emission spectrometry (ICP-AES) was used to preclude any significant interference from magnetic metals (Fe, Co, Ni, etc) on the reported magnetic behavior for CNS. All CNS samples prepared by different methods showed magnetic behavior, indicating that CNS has intrinsic magnetic property.

In fact, the twist of carbon materials to induce magnetism has been reported in the literature in recent years. Such as in 2010, N. Levy et al reported pseudo-magnetic fields greater than 300 Tesla induced by strain in graphene nanobubbles (see ref: *Science* **321**, 385-388 (2010)). Yuliang Li and coworkers reported a stable twisted 2D hydrocarbon tetraradical in 2016 (see ref: *Nat. Commun.* **7**, 1-11 (2016)). All these studies indicate that distortions between atomic layers in carbon material can generation of electrons out of the domain or free electrons, making them magnetic.

Figure B. HR-MS (MALDI-TOF) of CNS sample. The test result has verified that it does not contain any DDQ molecules.

In summary, the authors are encouraged to resubmit at a later stage, carefully addressing the 3D structural-magnetic property relationship of their system.

Response: We thank this reviewer very much for these helpful comments. In our

revised manuscript, we have performed spin-polarized density functional theory calculations. Computational results revealed that the magnetic property of CNS is derived from the moiety with the tension, as shown in the spin density of CNS (see Figures S29-S30 in our revised Supplementary Information). The related description (page15, in the first paragraph) has been revised as follows:

“...DFT calculation results show that the single-occupied molecular orbitals are composed of $2p$ orbital of C atoms and the whole spin density concentrates on the helix region (see Figures S29-S30). Therefore, it is concluded that magnetic property can be attributed to the breaking of π -type electrons induced by the strain region in the helix structures. Thus, it is possible that magnetic hysteresis under 10 K is attributed to the ferromagnetic coupling between two neighboring spins in the strain region of the helix. As the temperature increases, the hysteresis becomes smaller, probably due to the effect of spin-phonon interaction...”

Figure S29. Band structure with high-symmetry points (G: 0.0, 0.0, 0.0; Z: 0.0, 0.0, 0.5; F: 0.0, 0.5, 0.0; Q: 0.0, 0.5, 0.5) and density of states of CNS.

Figure S30. Spin density (a) and single-occupied molecular orbitals (b-c) of CNS.

Overall, we appreciate the reviewers very much for these comments and suggestions to help us improve our manuscript.

Reviewers' Comments:

Reviewer #1:

Remarks to the Author:

In the revised version, the authors tried to address a number of reviewers' concerns, improving the manuscript quality. Most of the elements of Fig. 1 are improved towards clarity. Interesting, one can see now structures closely resembling the ones experimentally (!) observed and fully simulated in 2011 J. Phys. Chem. Lett., 2, 2521-2524 by Y.Sun et al., Fig. 3c-d of carbon helices; they authors certainly should note this and cite that early paper. Although the authors tried to improve the characterization of their novel helicoids, this aspect remains not very strong. For instance, as Rev 2 pointed out, the pitch discord 2.5nm instead of 0.3-0.4 is not well explained, and quite large, may be due to intercalation or something else. Unfortunately in too many cases the authors cannot address the issues of *experimental* characterization and offer instead extra (and not really needed, due to prior theory-works) simulations and DFT analysis, not really asked for. The essential value of present report is in experimental findings and experimental evidence—and this remains a bit lacking. On the other hand I realize that the authors already reached the "saturation point" in their experimental characterization capacity. So the 'to be or not to be' question is now. I am inclined to recommend publication in present form (with very minor addition of ref. above; and also correct Fourier Transfer in lines 263 and 600 to F Transform), and see how the community responds. As the Rev. #2 says "fabrication of graphene nanosolenoids is a significant milestone" and this work, even though having a few loose ends, can motivate further efforts.

Reviewer #2:

Remarks to the Author:

Dear Editor

I am grateful to the authors for their efforts, and commend them on their report which attempts to fabricate carbon nanosolenoids (CNS). At this time, the experimental system remains challenging to characterize, and only non-conclusive data for successful CNS fabrication is available. Please find below few examples substantiating this statement.

"Although the model dimer or n-mer compounds can be used as a suitable reference compound to help analyze the structure of the CNS, it is regrettable that we cannot synthesize them within the limited time"

The authors did not have time to fabricate a model dimer or n-mer and solve its structure, which could have rendered their report, in principle, publishable after revising the inconsistencies in the CNS structural data and addressing potential statistical issues.

"Although the model dimer or n-mer compounds were not available, all these characterization data from NMR (1H MAS NMR, solid-state 13C NMR, 2D 1H-1H DQ-SQ NMR), Gel permeation chromatography, FT-IR spectrum, XPS spectrum, Raman spectrum, fluorescence spectrum, AFM images, HRTEM images and theoretical calculations can provide evidences to confirm that the CNS had been successfully prepared".

Except for few well-documented benchmark cases, neither chromatography, FT-IR, XPS, Raman, nor PL serve to resolve nanographene structures. NMR, AFM and HRTEM could provide evidence for a CNS structure, but in this case these techniques are inconclusive as the authors partly admit (below) and somewhat inconsistent (below), especially when the expected modelled CNS dimensions are still missing from the manuscript and possibly misleading (below).

"Following the reviewer's suggestion, the 3D molecular mechanics model (see Figure 1d in our revised manuscript and Figures S29-S30 in our revised Supplementary Information) and detailed AFM testing information have been provided in our revised manuscript"

There were previous concerns regarding the expected geometry and pitch of the CNS. The authors

needed to justify their previous claims of a 2.5 nm pitch and lead angle by means of a model. Figure 1d, S29-S30 does not appear to signal what are the modelled pitch, lead angles, etc. Moreover, a Riemann surface is a formal mathematical object with defined pitch, lead and dimensions which should be described in the manuscript and geometrically compared with the modelled CNS. The authors refer at one point to the "d-spacing" of 4.1 Å. It is not clear what this distance is, or if it accounts for the CNS sidechains. Intriguingly, Fig. 5b maintains the previous claim of a 2.5 nm pitch, while Fig. 5c highlights a distance of 4.0 Å corresponding to a "helical interlamellar spacing?".

"Furthermore, the HRTEM images in Figure S10 show clear lattice fringes corresponding to XRD patterns of self-assembly CNS, and the lattice fringe spacings are 2.65, 1.92 and 1.11 Å, respectively, which could be due to the self-assembly of CNS polymer dimers and trimers"

The authors appear to admit that the new XRD/HRTEM evidence for CNS is not conclusive.

Point-by-point response to reviewer's comments:

Reviewer #1

In the revised version, the authors tried to address a number of reviewers' concerns, improving the manuscript quality. Most of the elements of Fig. 1 are improved towards clarity. Interesting, one can see now structures closely resembling the ones experimentally (!) observed and fully simulated in **2011** *J. Phys. Chem. Lett.*, **2**, 2521-2524 by Y. Sun et al., Fig. 3c-d of carbon helices; they authors certainly should note this and cite that early paper.

Response: We thank this reviewer very much for these good comments and suggestions to help us improve our manuscript. As suggested, we have cited the reference [26] (*J. Phys. Chem. Lett.* **2**, 2521-2524 (2011)) in the Introduction section of the revised manuscript. And the related description (*page 4, in the first paragraph*) has also been revised as follows:

"...In three-dimensional (3D) graphene structures, new topology like helical spirals from graphite screw dislocations has been proposed.²⁵ Moreover, four kinds of dislocations and helical shapes were observed in raw anthracite using the bright-field high-resolution transmission electron microscopy (HRTEM).²⁶..."

Although the authors tried to improve the characterization of their novel helicoids, this aspect remains not very strong. For instance, as Rev 2 pointed out, the pitch discord 2.5nm instead of 0.3-0.4 is not well explained, and quite large, may be due to intercalation or something else.

Response: We thank this reviewer for pointing out this good comment. To elucidate the molecular structure of CNS helices, we further imaged it with low-dose integrated differential phase contrast scanning transmission electron microscopy (iDPC-STEM) techniques and realized atomic-scale high-resolution microstructure analysis (see Figure 5b in the revised manuscript). As a typical low-dose imaging technique, iDPC-STEM has been proved to be quite dose-efficient and sensitive to light elements (*Science* 2018, 359, 675-679; *Nat. Mater.* 2017, 16, 532-536; *Angew. Chem. Int. Ed.* 2020, 59, 819-825.), which is very suitable for high-resolution imaging of beam-sensitive organic materials like CNS helices. The aggregated CNS helices are well separated after prolonged probe sonication, which are then subject to low-dose iDPC-STEM imaging for resolving intrinsic single-stranded molecular structures. Specifically, Figure 5b-I shows the iDPC-STEM image of a single-stranded CNS helix with bright contrast that corresponds to the projected scalar electrostatic potentials of the molecule. The helical pitch and width of the CNS helix can be measured from the projected structure as ~0.4 nm and ~2.7 nm respectively, which match well with those values of the proposed CNS structural model upon a certain projection (~0.41 nm and ~2.48 nm when removing the side alkyl groups, see Figure 5b-III). In addition, the simulated projected potential map of a single-stranded CNS structural model embedded within a 1 nm-thick amorphous carbon layer also closely

resembles the high-resolution iDPC-STEM image (see Figure 5b-II). Unfortunately, we are still unable to further reasonably explain the pitch of 2.5 nm in the AFM image. To avoid confusion, we removed the AFM images and the corresponding descriptions from our revised manuscript. The related description and discussed (*page 12-14*) have been revised as follows:

“...Transmission electron microscopy (TEM) provides a powerful tool for visualizing the structures of organic molecules and polymers by direct imaging.^{50,51} It even allows the explicit elucidation of quite complicated spiral nanostructures.⁵² One major challenge for the direct TEM imaging of the **CNS** helix is the electron-beam irradiation induced structural damage. Unlike graphene with a two-dimensional covalent network and predominately subjected to the knock-on damage mechanism,⁵³ the beam damage mechanisms for helical **CNS** molecules are more complicated and may include considerable ionization effects. This means that simply lowering the accelerating voltage of electron beam that greatly eases the knock-on damage may not be able to alleviate the overall beam damage of **CNS** molecules. We evaluate the beam damage effects over **CNS** molecules using conventional TEM operated under a moderate accelerating voltage of 200 kV. Figure 5a shows the low-magnification TEM image covering a relatively large area of film assembled by **CNS** molecules. Careful inspection reveals that the **CNS** molecular strands assembled into domains with spring-like lattice fringes characterized for the helical structure (Figure 5a-I and Supplementary Figure 8), which probably originates from the magnetic interactions among **CNS** molecular strands. From an enlarged region of interest (Figure 5a-II), the helical pitch of ~ 4.1 Å can be clearly measured from the width of periodic lattice fringes of **CNS** strands, which corresponds to the spots in the fast Fourier transform (FFT) pattern (Figure 5a-III) from the white rectangle in Figure 5a-I. Notably, the packing order of **CNS** molecules is rarely observed by low-magnification TEM or scanning electron microscopy (Supplementary Figure 10), which is however visible from powder XRD pattern (Supplementary Figure 7) and high-magnification TEM images taken from small local regions (Supplementary Figure 9). Accordingly, the packing order of **CNS** assembly is most likely damaged by electron beam irradiation and further molecular structure damage is observed upon prolonged irradiation, which prohibit the explicit structural elucidation of single-stranded **CNS** helices.

It has been recently reported that low-dose electron microscopy provides an efficient solution for the atomic- or molecular-resolution imaging of beam-sensitive materials.⁵⁴ As a typical low-dose imaging technique, integrated differential phase contrast scanning transmission electron microscopy (iDPC-STEM) has been proved to be quite dose-efficient and sensitive to light elements,^{55,56} which is very suitable for high-resolution imaging of beam-sensitive organic materials like **CNS** helices. The severely aggregated **CNS** helices are well separated after prolonged probe sonication, which are then subject to low-dose iDPC-STEM imaging for resolving intrinsic single-stranded molecular structures (Figure 5b). Specifically, Figure 5b-I shows the iDPC-STEM image of a single-stranded **CNS** helix with bright contrast that corresponds to the projected scalar electrostatic potentials of the molecule. The helical

pitch and width of the **CNS** helix can be measured from the projected structure as ~ 0.4 nm and ~ 2.7 nm respectively, which match well with those values of the proposed **CNS** structural model upon a certain projection (~ 0.41 nm and ~ 2.48 nm when removing the sidechains, see Figure 5b-**III**). In addition, the simulated projected potential map of a single-stranded **CNS** structural model embedded within a 1 nm-thick amorphous carbon layer also closely resembles the high-resolution iDPC-STEM image (Figure 5b-**II**). Based on all above results, the molecular structure of the **CNS** helices is unambiguously determined by low-dose electron microscopy....”

Figure 5. HRTEM and iDPC-STEM characterizations of **CNS**. (a) **(I)** A HRTEM image showing the 2D helical self-assembled **CNS**. The experiment was carried out on JEM ARM-200F microscope operated at 200 kV. **(II)** An enlarged area HRTEM image of the white rectangle in **(I)**. **(III)** The FFT pattern from the white rectangle in **(I)**. (b) **(I)** Low-dose iDPC-STEM image showing the single strand **CNS**. **(II)** Simulated projected potential (amorphous carbon substrate) and **(III)** structural model of the **CNS**. A specific point-spread-function (PSF) width of 1.5 Å was used for **CNS**. Average measured helical pitch of ~ 0.4 nm and width of ~ 2.7 nm of **CNS**.

Unfortunately in too many cases the authors cannot address the issues of *experimental* characterization and offer instead extra (and not really needed, due to prior theory-works) simulations and DFT analysis, not really asked for. The essential value of present report is in experimental findings and experimental evidence-and this remains a bit lacking. On the other hand, I realize that the authors already reached the

“saturation point” in their experimental characterization capacity. So the ‘to be or not to be’ question is now. I am inclined to recommend publication in present form (with very minor addition of ref. above; and also correct Fourier Transfer in lines 263 nd 600 to F Transform), and see how the community responds. As the Rev. #2 says “fabrication of graphene nanosolenoids is a significant milestone” and this work, even though having a few loose ends, can motivate further efforts.

Response: We thank this reviewer very much for these good comments and suggestions to help us improve our manuscript. We agree with the reviewer that the essential value of the present study is in experimental findings and experimental evidence, and we are also committed to providing more sufficient experimental evidence. In our revised manuscript, we have further imaged **CNS** helices with low-dose iDPC-STEM techniques and realized atomic-scale high-resolution microstructure analysis (see Figure 5b in our revised manuscript). As a typical low-dose imaging technique, iDPC-STEM has been proved to be quite dose-efficient and sensitive to light elements, which is very suitable for high-resolution imaging of beam-sensitive organic materials like **CNS** helices. The severely aggregated **CNS** helices are well separated after prolonged probe sonication, which are then subject to low-dose iDPC-STEM imaging for resolving intrinsic single-stranded molecular structures. Specifically, Figure 5b-I shows the iDPC-STEM image of a single-stranded **CNS** helix with bright contrast that corresponds to the projected scalar electrostatic potentials of the molecule. The helical pitch and width of the **CNS** helix can be measured from the projected structure as ~ 0.4 nm and ~ 2.7 nm respectively, which match well with those values of the proposed **CNS** structural model upon a certain projection (~ 0.41 nm and ~ 2.48 nm when removing the side alkyl groups, see Figure 5b-III). In addition, the simulated projected potential map of a single-stranded **CNS** structural model embedded within a 1 nm-thick amorphous carbon layer also closely resembles the high-resolution iDPC-STEM image (see Figure 5b-II). The related description and discussed (*page 12-14*) have been revised as follows:

“...Transmission electron microscopy (TEM) provides a powerful tool for visualizing the structures of organic molecules and polymers by direct imaging.^{50,51} It even allows the explicit elucidation of quite complicated spiral nanostructures.⁵² One major challenge for the direct TEM imaging of the **CNS** helix is the electron-beam irradiation induced structural damage. Unlike graphene with a two-dimensional covalent network and predominately subjected to the knock-on damage mechanism,⁵³ the beam damage mechanisms for helical **CNS** molecules are more complicated and may include considerable ionization effects. This means that simply lowering the accelerating voltage of electron beam that greatly eases the knock-on damage may not be able to alleviate the overall beam damage of **CNS** molecules. We evaluate the beam damage effects over **CNS** molecules using conventional TEM operated under a moderate accelerating voltage of 200 kV. Figure 5a shows the low-magnification TEM image covering a relatively large area of film assembled by **CNS** molecules. Careful inspection reveals that the **CNS** molecular strands assembled into domains

with spring-like lattice fringes characterized for the helical structure (Figure 5a-I and Supplementary Figure 8), which probably originates from the magnetic interactions among **CNS** molecular strands. From an enlarged region of interest (Figure 5a-II), the helical pitch of ~ 4.1 Å can be clearly measured from the width of periodic lattice fringes of **CNS** strands, which corresponds to the spots in the fast Fourier transform (FFT) pattern (Figure 5a-III) from the white rectangle in Figure 5a-I. Notably, the packing order of **CNS** molecules is rarely observed by low-magnification TEM or scanning electron microscopy (Supplementary Figure 10), which is however visible from powder XRD pattern (Supplementary Figure 7) and high-magnification TEM images taken from small local regions (Supplementary Figure 9). Accordingly, the packing order of **CNS** assembly is most likely damaged by electron beam irradiation and further molecular structure damage is observed upon prolonged irradiation, which prohibit the explicit structural elucidation of single-stranded **CNS** helices.

It has been recently reported that low-dose electron microscopy provides an efficient solution for the atomic- or molecular-resolution imaging of beam-sensitive materials.⁵⁴ As a typical low-dose imaging technique, integrated differential phase contrast scanning transmission electron microscopy (iDPC-STEM) has been proved to be quite dose-efficient and sensitive to light elements,^{55, 56} which is very suitable for high-resolution imaging of beam-sensitive organic materials like **CNS** helices. The severely aggregated **CNS** helices are well separated after prolonged probe sonication, which are then subject to low-dose iDPC-STEM imaging for resolving intrinsic single-stranded molecular structures (Figure 5b). Specifically, Figure 5b-I shows the iDPC-STEM image of a single-stranded **CNS** helix with bright contrast that corresponds to the projected scalar electrostatic potentials of the molecule. The helical pitch and width of the **CNS** helix can be measured from the projected structure as ~ 0.4 nm and ~ 2.7 nm respectively, which match well with those values of the proposed **CNS** structural model upon a certain projection (~ 0.41 nm and ~ 2.48 nm when removing the sidechains, see Figure 5b-III). In addition, the simulated projected potential map of a single-stranded **CNS** structural model embedded within a 1 nm-thick amorphous carbon layer also closely resembles the high-resolution iDPC-STEM image (Figure 5b-II). Based on all above results, the molecular structure of the **CNS** helices is unambiguously determined by low-dose electron microscopy....”

As this reviewer pointed out, we have corrected Fast Fourier Transfer in lines 263 and 600 to fast Fourier transform (FFT) pattern in our revised manuscript. The corresponding parts (*the Structural Elucidation section and Caption to Figure 5*) have been revised as follows:

The Structural Elucidation section: “...From an enlarged region of interest (Figure 5a-II), the helical pitch of ~ 4.1 Å can be clearly measured from the width of periodic lattice fringes of **CNS** strands, which corresponds to the spots in the fast Fourier transform (FFT) pattern (Figure 5a-III) from the white rectangle in Figure 5a-I...”

Caption to Figure 5a: “...(III) The FFT pattern from the white rectangle in (I)...”

Overall, we thank this reviewer very much for these good comments and suggestions to help us improve our manuscript.

Reviewer #2

I am grateful to the authors for their efforts, and commend them on their report which attempts to fabricate carbon nanosolenoids (CNS). At this time, the experimental system remains challenging to characterize, and only non-conclusive data for successful CNS fabrication is available. Please find below few examples substantiating this statement.

Response: We thank this reviewer very much for these good suggestions to help us improve our manuscript. We agree with the reviewer that the experimental system remains challenging to characterize. As this reviewer pointed out, we are also committed to providing more sufficient experimental evidences and constantly improving our manuscripts.

“Although the model dimer or *n*-mer compounds can be used as a suitable reference compound to help analyze the structure of the CNS, it is regrettable that we cannot synthesize them within the limited time”

The authors did not have time to fabricate a model dimer or *n*-mer and solve its structure, which could have rendered their report, in principle, publishable after revising the inconsistencies in the CNS structural data and addressing potential statistical issues.

Response: We thank the reviewer for this good comment and suggestion. As suggested by this reviewer, it is a reasonable and effective strategy to further assist analyze the structures and properties of the target polymer by fabricating model dimer or *n*-mer compounds. This strategy has been reported in the study of planar conjugated polymers (for example: *Nat. Chem.*2014, **6**, 126-132), in which model dimer or *n*-mer compounds can be used as effective repeating units, i.e., the model dimer or *n*-mer compounds can show the structural characteristics of target polymer to some extent, but it may not be applicable for the helical nanographene material. The reasons are mainly summarized as three major points: 1) It is found that the model dimer or trimer of CNS still cannot form a complete helix layer. However, the structure and properties of CNS are affected by the stacking, extrusion and other interactions between the helix layers, which is also reflected in previous work of helically coiled carbon nanomaterials (see ref: *Angew. Chem. Int. Ed.* **56**, 6213-6217 (2017)). 2) Fabrication of twisted or helical graphene fragments requires a great deal of workload and is extremely challenging. A few cases reported previously were published as independent articles rather than as reference compounds for target polymers (see ref: *J. Am. Chem. Soc.* **140**, 4317-4326 (2018); *Angew. Chem. Int. Ed.* **56**, 3374-3378 (2017); *Angew. Chem. Int. Ed.* **57**, 5938-5942 (2018); *Angew. Chem. Int. Ed.* **57**, 6774-6779 (2018); *Angew. Chem. Int. Ed.* **57**, 14782-14786 (2018)). 3) Although no assistive analysis data for model dimer or *n*-mer compounds are available, we have further imaged CNS helices with low-dose iDPC-STEM techniques in our revised manuscript, which provides direct structural evidence for the

successful synthesis of target helical graphene materials. For the above reasons, we respectfully hope that this excellent reviewer will agree that we do not add model dimers or *n*-mer compounds as assistive analysis.

Following the reviewer's suggestion, to avoid confusion, we have removed the AFM images and the corresponding descriptions (the microscopic analysis section) which is difficult to be reasonably explained at present from our revised manuscript. In addition, to elucidate the molecular structure of **CNS** helices, we further imaged it with **low-dose integrated differential phase contrast scanning transmission electron microscopy (iDPC-STEM) techniques** and realized atomic-scale high-resolution microstructure analysis (see Figure 5b in our revised manuscript). As a typical low-dose imaging technique, iDPC-STEM has been proved to be quite dose-efficient and sensitive to light elements (*Science* 2018, 359, 675-679; *Nat. Mater.* 2017, 16, 532-536; *Angew. Chem. Int. Ed.* 2020, 59, 819-825.), which is very suitable for high-resolution imaging of beam-sensitive organic materials like **CNS** helices. The severely aggregated **CNS** helices are well separated after prolonged probe sonication, which are then subject to low-dose iDPC-STEM imaging for resolving intrinsic single-stranded molecular structures. Specifically, Figure 5b-I shows the iDPC-STEM image of a single-stranded **CNS** helix with bright contrast that corresponds to the projected scalar electrostatic potentials of the molecule. The helical pitch and width of the **CNS** helix can be measured from the projected structure as ~0.4 nm and ~2.7 nm respectively, which match well with those values of the proposed **CNS** structural model upon a certain projection (~0.41 nm and ~2.48 nm when removing the alkyl sidechains, see Figure 5b-III). In addition, the simulated projected potential map of a single-stranded **CNS** structural model embedded within a 1 nm-thick amorphous carbon layer also closely resembles the high-resolution iDPC-STEM image (see Figure 5b-II). The related figure, description and discussion (page 12-14) have been revised as follows:

Figure 5. HRTEM and iDPC-STEM characterizations of **CNS**. (a) **(I)** A HRTEM image showing the 2D helical self-assembled **CNS**. The experiment was carried out on JEM ARM-200F microscope operated at 200 kV. **(II)** An enlarged area HRTEM image of the white rectangle in **(I)**. **(III)** The FFT pattern from the white rectangle in **(I)**. (b) **(I)** Low-dose iDPC-STEM image showing the single strand **CNS**. **(II)** Simulated projected potential (amorphous carbon substrate) and **(III)** structural model of the **CNS**. A specific point-spread-function (PSF) width of 1.5 Å was used for **CNS**. Average measured helical pitch of ~0.4 nm and width of ~2.7 nm of **CNS**.

“...Transmission electron microscopy (TEM) provides a powerful tool for visualizing the structures of organic molecules and polymers by direct imaging.^{50,51} It even allows the explicit elucidation of quite complicated spiral nanostructures.⁵² One major challenge for the direct TEM imaging of the **CNS** helix is the electron-beam irradiation induced structural damage. Unlike graphene with a two-dimensional covalent network and predominately subjected to the knock-on damage mechanism,⁵³ the beam damage mechanisms for helical **CNS** molecules are more complicated and may include considerable ionization effects. This means that simply lowering the accelerating voltage of electron beam that greatly eases the knock-on damage may not be able to alleviate the overall beam damage of **CNS** molecules. We evaluate the beam damage effects over **CNS** molecules using conventional TEM operated under a moderate accelerating voltage of 200 kV. Figure 5a shows the low-magnification TEM image covering a relatively large area of film assembled by **CNS** molecules. Careful inspection reveals that the **CNS** molecular strands assembled into domains with spring-like lattice fringes characterized for the helical structure (Figure 5a-**I** and Supplementary Figure 8), which probably originates from the magnetic interactions among **CNS** molecular strands. From an enlarged region of interest (Figure 5a-**II**), the

helical pitch of ~ 4.1 Å can be clearly measured from the width of periodic lattice fringes of **CNS** strands, which corresponds to the spots in the fast Fourier transform (FFT) pattern (Figure 5a-III) from the white rectangle in Figure 5a-I. Notably, the packing order of **CNS** molecules is rarely observed by low-magnification TEM or scanning electron microscopy (Supplementary Figure 10), which is however visible from powder XRD pattern (Supplementary Figure 7) and high-magnification TEM images taken from small local regions (Supplementary Figure 9). Accordingly, the packing order of **CNS** assembly is most likely damaged by electron beam irradiation and further molecular structure damage is observed upon prolonged irradiation, which prohibit the explicit structural elucidation of single-stranded **CNS** helices.

It has been recently reported that low-dose electron microscopy provides an efficient solution for the atomic- or molecular-resolution imaging of beam-sensitive materials.⁵⁴ As a typical low-dose imaging technique, integrated differential phase contrast scanning transmission electron microscopy (iDPC-STEM) has been proved to be quite dose-efficient and sensitive to light elements,^{55,56} which is very suitable for high-resolution imaging of beam-sensitive organic materials like **CNS** helices. The severely aggregated **CNS** helices are well separated after prolonged probe sonication, which are then subject to low-dose iDPC-STEM imaging for resolving intrinsic single-stranded molecular structures (Figure 5b). Specifically, Figure 5b-I shows the iDPC-STEM image of a single-stranded **CNS** helix with bright contrast that corresponds to the projected scalar electrostatic potentials of the molecule. The helical pitch and width of the **CNS** helix can be measured from the projected structure as ~ 0.4 nm and ~ 2.7 nm respectively, which match well with those values of the proposed **CNS** structural model upon a certain projection (~ 0.41 nm and ~ 2.48 nm when removing the sidechains, see Figure 5b-III). In addition, the simulated projected potential map of a single-stranded **CNS** structural model embedded within a 1 nm-thick amorphous carbon layer also closely resembles the high-resolution iDPC-STEM image (Figure 5b-II). Based on all above results, the molecular structure of the **CNS** helices is unambiguously determined by low-dose electron microscopy....”

“Although the model dimer or *n*-mer compounds were not available, all these characterization data from NMR (¹H MAS NMR, solid-state ¹³C NMR, 2D ¹H-¹H DQ-SQ NMR), Gel permeation chromatography, 13 FT-IR spectrum, XPS spectrum, Raman spectrum, fluorescence spectrum, AFM images, HRTEM images and theoretical calculations can provide evidences to confirm that the **CNS** had been successfully prepared”

Except for few well-documented benchmark cases, neither chromatography, FT-IR, XPS, Raman, nor PL serve to resolve nanographene structures. NMR, AFM and HRTEM could provide evidence for a **CNS** structure, but in this case these techniques are inconclusive as the authors partly admit (below) and somewhat inconsistent (below), especially when the expected modelled **CNS** dimensions are still missing from the manuscript and possibly misleading (below).

Response: We thank this reviewer very much for these good comments and

suggestions to help us improve our manuscript. In our revised manuscript, to elucidate the molecular structure of **CNS** helices, we further imaged it with low-dose integrated differential phase contrast scanning transmission electron microscopy (iDPC-STEM) techniques and realized atomic-scale high-resolution microstructure analysis (see Figure 5b in our revised manuscript). As a typical low-dose imaging technique, iDPC-STEM has been proved to be quite dose-efficient and sensitive to light elements, which is very suitable for high-resolution imaging of beam-sensitive organic materials like **CNS** helices. The severely aggregated **CNS** helices are well separated after prolonged probe sonication, which are then subject to low-dose iDPC-STEM imaging for resolving intrinsic single-stranded molecular structures. Specifically, Figure 5b-I shows the iDPC-STEM image of a single-stranded **CNS** helix with bright contrast that corresponds to the projected scalar electrostatic potentials of the molecule. The helical pitch and width of the **CNS** helix can be measured from the projected structure as ~ 0.4 nm and ~ 2.7 nm respectively, which match well with those values of the proposed **CNS** structural model upon a certain projection (~ 0.41 nm and ~ 2.48 nm when removing the alkyl sidechains, see Figure 5b-III). In addition, the simulated projected potential map of a single-stranded **CNS** structural model embedded within a 1 nm-thick amorphous carbon layer also closely resembles the high-resolution iDPC-STEM image (see Figure 5b-II).

In addition, we have also marked the molecular dimensions modeling information of **CNS** in Figure 1d-III, and the related description (*page 6-7*) has been revised as follows:

“...To investigate atomic structure information such as diameter, helical pitch, edges, helical core, coil angle, stacking, band, spin density, molecular orbitals and so on, density-functional theory (DFT) calculations were carried out for the geometric structure and electronic structure of **CNS** using the Vienna an initio simulation package (VASP). The results are summarized in Figure 1d (II and III), Supplementary Figures 28-29 and Supplementary Table 1. Figure 1d-II shows the geometrical optimization structure of **CNS**, and the results indicate that **CNS** features helix-bundles configuration along the Burgers vector **b** (parallel to the *c*-axis marked in purple, see Figure 1d-III) with the outer diameter, helical pitch and coil angle of $D = 24.782$ Å, $p = 4.118$ Å and $\gamma = 3.028^\circ$ [$\gamma = \arctan(p/(\pi D))$], respectively.²⁵...”

Figure 1d-III. (III) Helical structure of CNS without the alkyl substituent groups. Burgers vector **b** parallel to the *c*-axis (purple). *D*, *p*, and γ are the outer diameter, pitch, and coil angle, respectively.

“Following the reviewer’s suggestion, the 3D molecular mechanics model (see Figure 1d in our revised manuscript and Figures S29-S30 in our revised Supplementary Information) and detailed AFM testing information have been provided in our revised manuscript”

There were previous concerns regarding the expected geometry and pitch of the CNS. The authors needed to justify their previous claims of a 2.5 nm pitch and lead angle by means of a model. Figure 1d, S29-S30 does not appear to signal what are the modelled pitch, lead angles, etc. Moreover, a Riemann surface is a formal mathematical object with defined pitch, lead and dimensions which should be described in the manuscript and geometrically compared with the modelled CNS. The authors refer at one point to the “d-spacing” of 4.1 Å. It is not clear what this distance is, or if it accounts for the CNS sidechains. Intriguingly, Fig. 5b maintains the previous claim of a 2.5 nm pitch, while Fig. 5c highlights a distance of 4.0 Å corresponding to a “helical interlamellar spacing?”

Response: We thank this reviewer for pointing out this. To elucidate the molecular structure of CNS helices, we further imaged it with low-dose integrated differential phase contrast scanning transmission electron microscopy (iDPC-STEM) techniques and realized atomic-scale high-resolution microstructure analysis (see Figure 5b in our revised manuscript). As a typical low-dose imaging technique, iDPC-STEM has been proved to be quite dose-efficient and sensitive to light elements, which is very suitable for high-resolution imaging of beam-sensitive organic materials like CNS helices. The severely aggregated CNS helices are well separated after prolonged probe sonication, which are then subject to low-dose iDPC-STEM imaging for resolving intrinsic single-stranded molecular structures. Specifically, Figure 5b-I shows the iDPC-STEM image of a single-stranded CNS helix with bright contrast that corresponds to the projected scalar electrostatic potentials of the molecule. The helical pitch and width of the CNS helix can be measured from the projected structure as ~0.4 nm and ~2.7 nm respectively, which match well with those values of the

proposed **CNS** structural model upon a certain projection (~ 0.41 nm and ~ 2.48 nm when removing the alkyl sidechains, see Figure 5b-**III**). In addition, the simulated projected potential map of a single-stranded **CNS** structural model embedded within a 1 nm-thick amorphous carbon layer also closely resembles the high-resolution iDPC-STEM image (see Figure 5b-**II**). Unfortunately, we are still unable to further reasonably explain the pitch of 2.5 nm in the AFM image. To avoid confusion, we have already removed the AFM images and the corresponding descriptions from our revised manuscript. In addition, we have also marked the molecular dimensions modeling information, including the outer diameter, helical pitch, and coil angle of **CNS** in Figure 1d-**III**.

“Furthermore, the HRTEM images in Figure S10 show clear lattice fringes corresponding to XRD patterns of self-assembly **CNS**, and the lattice fringe spacings are 2.65, 1.92 and 1.11 Å, respectively, which could be due to the self-assembly of **CNS** polymer dimers and trimers”

The authors appear to admit that the new XRD/HRTEM evidence for **CNS** is not conclusive.

Response: We thank this reviewer for pointing out this. To elucidate the molecular structure of **CNS** helices, we further imaged it with low-dose integrated differential phase contrast scanning transmission electron microscopy (iDPC-STEM) techniques and realized atomic-scale high-resolution microstructure analysis (see Figure 5b in our revised manuscript). As a typical low-dose imaging technique, iDPC-STEM has been proved to be quite dose-efficient and sensitive to light elements, which is very suitable for high-resolution imaging of beam-sensitive organic materials like **CNS** helices. The severely aggregated **CNS** helices are well separated after prolonged probe sonication, which are then subject to low-dose iDPC-STEM imaging for resolving intrinsic single-stranded molecular structures. Specifically, Figure 5b-**I** shows the iDPC-STEM image of a single-stranded **CNS** helix with bright contrast that corresponds to the projected scalar electrostatic potentials of the molecule. The helical pitch and width of the **CNS** helix can be measured from the projected structure as ~ 0.4 nm and ~ 2.7 nm respectively, which match well with those values of the proposed **CNS** structural model upon a certain projection (~ 0.41 nm and ~ 2.48 nm when removal of the sidechains, see Figure 5b-**III**). In addition, the simulated projected potential map of a single-stranded **CNS** structural model embedded within a 1 nm-thick amorphous carbon layer also closely resembles the high-resolution iDPC-STEM image (see Figure 5b-**II**).

Overall, we thank this reviewer very much for these good comments and suggestions to help us improve our manuscript.

Reviewers' Comments:

Reviewer #1:

Remarks to the Author:

In this version the authors tried, once again, to address the referees' concerns, to my opinion to a degree/way sufficient for accepting paper to Nat Comm. I notice that the title of [25] already made connection to Riemann Surfaces directly, so the authors should mention it, in spirit of good scholarship, and to avoid ambiguities in the future on 'who, what and when'... Even though in some places further experimental verification/data is highly desirable but very hard to obtain, publication at this time will be informative and stimulating for further research in the field, of clear interest for the broad readership of N Comm.

Reviewer #2:

Remarks to the Author:

The authors have revised their structural claims, and now interpret a helical pitch of 0.4 nm from iDPC-STEM imaging data in potential agreement with HRTEM data, instead of a 2.5 nm pitch previously observed by AFM. Moreover, they have included a DFT model and new structural data, Fig. 5.

The interpretation of the new and previous structural data in Fig. 5 is challenging due to the absence of key structural indicators in the Figures (e.g. the DFT $p=0.41$ nm and $D=2.48$ and experimental $p=0.41$ nm and $D=2.7$ pitch are not explicitly shown) and apparent absence of reproducible data sets. Moreover, the DFT methods appear to be missing. It is not known if the authors are employing (empirically) vdW-corrected DFT or otherwise. The detailed dimensions and structural indicators for one molecular model should be shown in a Figure and employed throughout (there are three or four different model CNS shown in Fig.1).

In view of the above, and the ambiguous structural proof in Fig. 5, the manuscript does not substantiate its claim of 'A Single-Stranded Magnetic Carbon Nanosolenoid with Riemann Surfaces'. A single-strand of a nanosolenoid is not recognizable in Fig. 5, neither is a Riemann surface which could have been argued with a model compound. To this Reviewer, there is no obvious "Y-form" with corresponding dimensions/indicators in Fig. 5 (but maybe in Supplementary Fig. 9, see below) which could have substantiated a claim relevant to the synthesis of dislocations or artificial Anthracite following Reviewer #1 comments.

In the absence of a structural model dimer or n-mer, the claim of the manuscript in its current form appears to be based on the expectation of the outcome of a polymerization and dehydrogenation hypothesis and not on experimental data. This is especially relevant since this Reviewer is not aware of a polymerization and dehydrogenation procedure of the reported synthesis, which has been structurally characterized. It is worth noting that to the best of this Reviewer's knowledge, unambiguous structural data from well-known phenylene polymers by the Fischer and Sinitskii groups, has mostly evidenced ~ 10 nm long, low-molecular weight linear graphene nanoribbons. Longer materials could still consist of a mixture of rigid graphene ribbons and flexible polymers. That is to say, the rational synthesis of well-known graphenic materials is still in its infancy, so that new protocols and materials should be stringently characterized.

That said, because lamellar "Y-forms" appear to be present in Supporting Fig. 9a, and Fig. 5, Supplementary Fig. 8 and Supplementary Fig. 9 evidence rich structural phases, this Reviewer could in principle recommend publication of this work following a revised claim. Such claim should not be based on the biased assumption of polymerization and cyclodehydrogenation of the desired structure, but rather on data analysis. By e.g. expanding the TEM/SEM analysis to the potential conformational space of polymer P1 and cross-correlating with TEM data, the authors might identify the products of their method. If the authors do reproducibly identify few consecutive "Y-forms", they may claim 'Identification of carbon nanosolenoids in the rational synthesis of graphitic polymers' in addition to side products, but otherwise the authors have a unique opportunity to revise their hypothesis based on their extensive TEM/SEM data set. The authors can also turn to identification of DDQ/TfOH-induced dehydrogenation defects, sonication defects, or electron-

irradiation defects, as observed in simple graphene nanoribbons (see kinks and defects in Suppl. Fig. 8 of Sinitzkii Nature Commun. 3189 2014 for instance). If the authors choose to revise their claims when processing their extensive TEM/SEM data set, it might be instrumental to present all quantifiers and extensive statistical analysis with appropriate error bars directly in the Figures.

Point-by-point response to reviewer's comments:

Reviewer #1

In this version the authors tried, once again, to address the referees' concerns, to my opinion to a degree/way sufficient for accepting paper to Nat Comm.

I notice that the title of [25] already made connection to Riemann Surfaces directly, so the authors should mention it, in spirit of good scholarship, and to avoid ambiguities in the future on 'who, what and when'... Even though in some places further experimental verification/data is highly desirable but very hard to obtain, publication at this time will be informative and stimulating for further research in the field, of clear interest for the broad readership of N Comm.

Response: We thank the reviewer for this good comment and nice suggestions. To avoid confusion, we have revised the corresponding description of the reference [25] (*Nano Lett.* **16**, 34-39 (2016)) in this revised manuscript to make connection to Riemann Surfaces directly. The related description (Introduction section, *the last sentence of the second paragraph*) has been revised as follows:

“...In 2016, Yakobson and coworkers initially predicted that a carbon solenoid with Riemann surfaces and small diameter can behave as a quantum conductor when a voltage is applied, resulting in a large magnetic field near the center and bringing about excellent inductance...”

Overall, we thank this reviewer very much for these good comments and suggestions to help us improve our manuscript.

Reviewer #2

To Reviewer #2: Since our submission to this journal, one year and two months have been passed. We fully appreciate all the comments and the valuable time spent by this reviewer. In addition, we wish the reviewer can understand how difficult to synthesize some impossible oligomers and how hard to do characterizations for this 3D polymeric material. At this time, we repeated the **iDPC-STEM and HRTEM tests** on different batches of CNS samples. The results show that the CNS samples have good reproducibility. As Reviewer #1 mentioned “Even though in some places further experimental verification/data is highly desirable but very hard to obtain, publication at this time will be informative and stimulating for further research in the field”. We really hope this reviewer can agree with us with these characterizations and we have done our best (these characterizations are very hard. Less than three groups can do iDPC-STEM over the world).

The authors have revised their structural claims, and now interpret a helical pitch of 0.4 nm from iDPC-STEM imaging data in potential agreement with HRTEM data, instead of a 2.5 nm pitch previously observed by AFM. Moreover, they have included a DFT model and new structural data, Fig. 5. The interpretation of the new and previous structural data in Fig. 5 is challenging due to the absence of key structural indicators in the Figures (e.g. the DFT $p = 0.41$ nm and $D = 2.48$ and experimental $p = 0.41$ nm and $D = 2.7$ pitch are not explicitly shown) and apparent absence of reproducible data sets.

Response: We thank this reviewer very much for pointing out this to help us improve our manuscript. Following the reviewer’s suggestion, the key structural indicators (e.g. the DFT of $p = 0.41$ nm and $D = 2.88$ nm and experimental of $p' = 0.4$ nm and $D' = 2.7$ nm) have been provided in Figure 5b of our revised manuscript. The related description (Structural Elucidation section, *the second paragraph*) and Figure 5b have been revised as follows:

“...The experimental helical pitch (p') and width (D') of the CNS helix can be statistical measured from the projected structure as ~ 0.4 nm and ~ 2.7 nm respectively (Figure 5b-II and Supplementary Figure 10), which match well with those values of the proposed CNS structural model upon a certain projection (when the alkyl sidechains were not removed, p and D were ~ 0.41 nm and ~ 2.88 nm respectively, see Figure 5b-III)...”

Figure 5b. (I) Low-dose iDPC-STEM image showing the single strand CNS. (II) Simulated projected potential (amorphous carbon substrate) of the CNS. A specific point-spread-function (PSF) width of 1.5 Å was used for CNS. Statistical analysis of CNS samples shows that measured helical pitch (p') and width (D') values are mainly distributed at ~ 0.4 nm and ~ 2.7 nm, respectively. (III) structural model of the CNS. Statistical analysis shows that the helical pitch (p) and width (D) calculated by DFT were ~ 0.41 nm and ~ 2.88 nm, respectively (with alkyl sidechains).

In order to verify the repeatability of CNS samples for preparation and structural elucidation, and to make statistical measurements of experimental structural indicators of CNS, **additional iDPC-STEM tests** were performed on different batches of CNS samples. The results of extensive iDPC-STEM tests were summarized in Supplementary Figure 10 in our revised manuscript. These results show that the CNS samples have good reproducibility for preparation and structural elucidation, and statistical analysis shows that the measured helical pitch (p') and width (D') of CNS are mainly concentrated at 0.4 nm and 2.7 nm, respectively. In addition, additional HRTEM tests on different batches of CNS samples (see Figures A and B) showed morphologies and lattice spacing similar to previous images, which further confirmed that CNS samples had reproducible data sets for preparation and structural elucidation.

Supplementary Figure 10. (a) Low-dose iDPC-STEM images with different fields of view of CNS samples from different batches. Scale bars: 1 nm. Statistical analysis of helical pitch length (b) and width (c) of CNS samples shows that measured helical pitch length and width values are mainly distributed at ~ 0.4 nm and ~ 2.7 nm, respectively.

Figure A. Spherical aberration-corrected HRTEM images with different fields of view and different resolutions of CNS. The experiment was carried out on JEM ARM-200F microscope operated at 200 kV.

Figure B. HRTEM images with different fields of view and different resolutions of CNS. The experiment was carried out on JEM-2100F microscope operated at 200 kV.

Moreover, the DFT methods appear to be missing. It is not known if the authors are employing (empirically) vdW-corrected DFT or otherwise. The detailed dimensions and structural indicators for one molecular model should be shown in a Figure and employed throughout (there are three or four different model CNS shown in Fig.1).

Response: We thank this reviewer very much for this point. In fact, we have described the DFT methods in our initial manuscript (see Computational Details

section in Supplementary Information). Notably, by default, van der Waals (vdW)-corrected DFT is usually used for Vienna an initio simulation package (VASP). DFT-D3 method with Becke-Jonson damping was used to correct vdW interactions resulting from dynamical correlations between fluctuating charge distributions. To avoid confusion, we have revised the corresponding description of the vdW-corrected DFT in our revised manuscript. The related description (Computational Details section in Supplementary Information) has been revised as follows:

“...A DFT-D3 method with Becke-Jonson damping^[S13] for dispersion correction was used to correct the van der Waals (vdW) interactions...”

In addition, following the reviewer’s suggestion, the detailed dimensions and structural indicators for CNS molecular model have been provided in Figure 5b-III in our revised manuscript.

Figure 5b. (I) Low-dose iDPC-STEM image showing the single strand CNS. (II) Simulated projected potential (amorphous carbon substrate) of the CNS. A specific point-spread-function (PSF) width of 1.5 Å was used for CNS. Statistical analysis of CNS samples shows that measured helical pitch (p') and width (D') values are mainly distributed at ~0.4 nm and ~2.7 nm, respectively. (III) structural model of the CNS. Statistical analysis shows that the helical pitch (p) and width (D) calculated by DFT were ~0.41 nm and ~2.88 nm, respectively (with alkyl sidechains).

In view of the above, and the ambiguous structural proof in Fig. 5, the manuscript does not substantiate its claim of ‘A Single-Stranded Magnetic Carbon Nanosolenoid with Riemann Surfaces’. A single-strand of a nanosolenoid is not recognizable in Fig. 5, neither is a Riemann surface which could have been argued with a model compound. To this Reviewer, there is no obvious “Y-form” with corresponding dimensions/indicators in Fig. 5 (but maybe in Supplementary Fig. 9, see below) which could have substantiated a claim relevant to the synthesis of dislocations or artificial Anthracite following Reviewer #1 comments.

Response: We appreciate for this comment but we cannot fully agree with these points and conclusions of this reviewer.

First of all, **there is no “Y-form” structure in our material samples.** The so-called “Y” observed in our HRTEM and iDPC-STEM is actually a projection of the helix topology that has the characteristics of continuous “Y” end-to-end and is highly

consistent with our structural model (see Figure 5b). However, the “Y-form” in anthracite emphasized by this reviewer is defined as “react by radical addition to the aromatic rings of a parallel graphenic layer” (see the following Figure C from ref *J. Phys. Chem. Lett.* **2**, 2521-2524 (2011)). It needs to be clearly pointed out that “Y-form” in anthracite has no the characteristics of continuous “Y” end-to-end (see the following Figure C-b' and Figure C-b''), which is obviously different from our CNS samples. In the light of the foregoing, we believe that this reviewer’s comment confuses the concept.

Figure C. Edge-radical sites bind to the adjacent layers resulting into a Y-form (b), containing sp^3 -carbon atoms, marked blue in (b''), and a corresponding TEM simulation (b'). Reprinted from ref *J. Phys. Chem. Lett.* **2**, 2521-2524 (2011).

Secondly, we do not agree with the reviewer’s comment of “the ambiguous structural proof in Fig. 5”. It is important to note that Figure 5b clearly shows the single-stranded structure of the target compound and is well matched with the proposed CNS structural model. It is well-known that using optimized structural models to match the obtained electron microscope images to determine the correctness of emerging topologies is the most common and effective structural confirmation method (see ref: *Nature* **2010**, 466, 470-473; *J. Am. Chem. Soc.* **2015**, 137, 6097-6103; *J. Am. Chem. Soc.* **2015**, 137, 4022-4025; *J. Am. Chem. Soc.* **2015**, 137, 1802-1808; *Chem. Soc. Rev.*, **2015**, 44, 6616-6643; *J. Am. Chem. Soc.* **2017**, 139, 8698-8704; *J. Am. Chem. Soc.* **2020**, 142, 13162-13169). Of course, in order to avoid confusion, we have removed the “single-stranded” description of the corresponding sections (Title and Discussion) in our revised manuscript.

Finally, we designed and synthesized nanoscale entity similar to this predicted by the reference of *Nano Lett.* (*Nano Lett.* **16**, 34-39 (2016)). Although it is difficult to identify the target polymer’s Riemannian surface at the atomic level, combined with

the DFT simulation (the DFT calculations details can be found in the “Computational Details” section of Supplementary Information, and the corresponding explanations and discussions appear mainly in the “Molecular Design and Theoretical Calculations of CNS” section of our revised manuscript and are used throughout the whole manuscript), we can explain the reasonableness of our structural design, as well as the special structures-properties (such as the magnetic properties originally envisaged due to high distortion) relationship that have also been demonstrated experimentally.

In summary, while we appreciate the reviewer’s comments to help us improve our manuscript, we respectfully hope that the reviewer will not overlook the difficulties in our experimental measurements, as did Reviewer #1 comments “Even though in some places further experimental verification/data is desirable but very hard to obtain”.

In the absence of a structural model dimer or n-mer, the claim of the manuscript in its current form appears to be based on the expectation of the outcome of a polymerization and dehydrogenation hypothesis and not on experimental data. This is especially relevant since this Reviewer is not aware of a polymerization and dehydrogenation procedure of the reported synthesis, which has been structurally characterized. It is worth noting that to the best of this Reviewer’s knowledge, unambiguous structural data from well-known phenylene polymers by the Fischer and Sinitskii groups, has mostly evidenced ~10 nm long, low-molecular weight linear graphene nanoribbons. Longer materials could still consist of a mixture of rigid graphene ribbons and flexible polymers. That is to say, the rational synthesis of well-known graphenic materials is still in its infancy, so that new protocols and materials should be stringently characterized. That said, because lamellar “Y-forms” appear to be present in Supporting Fig. 9a, and Fig. 5, Supplementary Fig. 8 and Supplementary Fig. 9 evidence rich structural phases, this Reviewer could in principle recommend publication of this work following a revised claim. Such claim should not be based on the biased assumption of polymerization and cyclodehydrogenation of the desired structure, but rather on data analysis. By e.g. expanding the TEM/SEM analysis to the potential conformational space of polymer P1 and cross-correlating with TEM data, the authors might identify the products of their method. If the authors do reproducibly identify few consecutive “Y-forms”, they may claim ‘Identification of carbon nanosolenoids in the rational synthesis of graphitic polymers’ in addition to side products, but otherwise the authors have a unique opportunity to revise their hypothesis based on their extensive TEM/SEM data set. The authors can also turn to identification of DDQ/TfOH-induced dehydrogenation defects, sonication defects, or electron-irradiation defects, as observed in simple graphene nanoribbons (see kinks and defects in Suppl. Fig. 8 of Sinitskii Nature Commun. 3189 2014 for instance). If the authors choose to revise their claims when processing their extensive TEM/SEM data set, it might be instrumental to present all quantifiers and extensive statistical analysis with appropriate error bars directly in the Figures.

Response: We thank this reviewer very much for his/her comments to help us improve our manuscript.

There is no “Y-form” structure in our material samples. The so-called “Y” observed in our HRTEM and iDPC-STEM is actually a projection of the helix topology that has the characteristics of continuous “Y” end-to-end and is highly consistent with our structural model (see Figure 5b). However, the “Y-form” in anthracite emphasized by this reviewer is defined as “react by radical addition to the aromatic rings of a parallel graphenic layer” (see Figure C and ref *J. Phys. Chem. Lett.* **2**, 2521-2524 (2011)). Both are very different. The latter is a stochastic material defect and has no “Y” head to tail continuity. In the light of the foregoing, we believe that this reviewer’s comment confuses the concept.

In addition, the characterization for defects exemplified by this reviewer all comes from the reported planar materials (such as graphene, graphene nanoribbons), which should be much easier. However, **the characterization of magnetic nanoscale non-planar molecules/polymers similar to CNS was not previously reported**, making the characterization of such materials challenging for the following reasons:

One major challenge for the direct TEM imaging of the CNS helix is the electron-beam irradiation induced structural damage. Unlike graphene with a two-dimensional covalent network and predominately subjected to the knock-on damage mechanism, the beam damage mechanisms for this 3D CNS molecules are more complicated and have considerable ionization effects. This means that simply lowering the accelerating voltage of electron beam that greatly eases the knock-on damage may not be able to alleviate the overall beam damage of CNS molecules.

Originating from the magnetic interactions among CNS molecular strands, CNS molecular strands may assemble into domains with spring-like lattice fringes characterized for the helical structure (Figure 5a-I and Supplementary Figure 8 in our initial manuscript).

Besides, the packing order of CNS assembly is most likely damaged by electron beam irradiation and further molecular structure damage is observed upon prolonged irradiation, which prohibit the explicit structural elucidation of single-stranded CNS helices.

To sum up, **additional iDPC-STEM and HRTEM tests** were performed on different batches of CNS samples. The results show that the CNS samples have good reproducibility for preparation and structural elucidation, and statistical analysis shows that the measured helical pitch (p') and width (D') of CNS are mainly concentrated at 0.40 nm and 2.7 nm, respectively, which further confirmed that CNS samples had reproducible data sets for preparation and structural elucidation. In addition, structural damage caused by electron-beam irradiation and magnetic agglomeration caused by the magnetism of CNS makes the structural elucidation of CNS much more challenging and difficult than planar graphene-based materials.

Overall, we thank this reviewer very much for these good comments and suggestions to help us improve our manuscript.

Reviewers' Comments:

Reviewer #2:

None

Reviewer #3:

Remarks to the Author:

This manuscript by Du and coworkers describes the synthesis and characterization of a magnetic carbon nanosolenoid with Riemann surfaces. The manuscript was submitted and revised once following reviewers' comments. After the second submission, reviewer #2 still expressed doubt regarding the structural integrity of the structure. Although I acknowledge that having irrefutable proof of such structures is almost impossible, I think that the authors did everything they could to get as much proof as possible. What they get in low-dose iDOC-STEM is for me quite conclusive and these results are more than enough to warrant publication in NatureComm. These are very high quality results that will have an impact in the carbon nano materials community. Thus, I recommend this version of the manuscript to be published in NatureComm.

FOURTH ROUND REVIEWERS' COMMENTS

Reviewer #3 (Remarks to the Author):

This manuscript by Du and coworkers describes the synthesis and characterization of a magnetic carbon nanosolenoid with Riemann surfaces. The manuscript was submitted and revised once following reviewers' comments. After the second submission, reviewer #2 still expressed doubt regarding the structural integrity of the structure. Although I acknowledge that having irrefutable proof of such structures is almost impossible, I think that the authors did everything they could to get as much proof as possible. What they get in low-dose iDOC-STEM is for me quite conclusive and these results are more than enough to warrant publication in NatureComm. These are very high quality results that will have an impact in the carbon nano materials community. Thus, I recommend this version of the manuscript to be published in NatureComm.

***Response:** We thank this reviewer very much for these helpful comments to publish our manuscript.*